# ENLIGHTENMENT PERIOD IMPROVING DNN PERFOR-MANCE

## ABSTRACT

The start of deep neural network training is characterized by a brief yet critical phase that lasts from the beginning of the training until the accuracy reaches approximately 50%. During this phase, disordered representations rapidly transition toward ordered structure, and we term this phase the Enlightenment Period. Through theoretical modeling based on phase transition theory and experimental validation, we reveal that applying Mixup data augmentation during this phase has a dual effect: it introduces a Gradient Interference Effect that hinders performance, while also providing a beneficial Activation Revival Effect to restore gradient updates for saturated neurons. We further demonstrate that this negative interference diminishes as the sample set size or the model parameter size increases, thereby shifting the balance between these two effects. Based on these findings, we propose three strategies that improve performance by solely adjusting the training data distribution within this brief period: the Mixup Pause Strategy for small-scale scenarios, the Alpha Boost Strategy for large-scale scenarios with underfitting, and the High-Loss Removal Strategy for tasks where Mixup is inapplicable (e.g., time series and large language models). Extensive experiments show that these strategies achieve superior performance across diverse architectures such as ViT and ResNet on datasets including CIFAR and ImageNet-1K. Ultimately, this work offers a novel perspective on enhancing model performance by strategically capitalizing on the dynamics of the brief and crucial early stages of training. Code is available at https://anonymous.4open.science/r/code-A5F1/.

## 1 INTRODUCTION

The early stage of neural network training is characterized by parameter update fluctuations, a sharp drop in loss values, and the rapid formation of discriminative representations. After this short yet critical phase, all training curves flatten and stabilize until convergence. Although numerous studies(Hu et al., 2020; Kleinman et al., 2023; Koch & Ghosh, 2025; Paul et al., 2021) have attempted to understand these early dynamic processes, the strong nonlinearity governing the gradient flow and non-convexity of the parameter space make constructing a closed-form solution highly challenging.

Figure 1(light gray background area, first 20 epochs) shows that accuracy and two metrics measuring parameter dynamics, namely the Batch-Epoch Normalized Ratio (BENR) and the Activation Trajectory Distance (ATD) all exhibit severe fluctuations (see Appendix C). In both vanilla training and Input Mixup, hierarchical feature distributions evolve from nearly disorganized clusters to compact ones with clear class boundaries, accompanied by a simultaneous improvement in classification accuracy.

Thus, it is not only a period of significant changes in the parameters or representations but also a phase in which the core representations of the model form. In Figure 1, although Mixup yields clearer class boundaries when training converges, its early embedding results are much more disorganized. This indicates that the evolutionary paths of vanilla training and Mixup training diverge starting from the initial epochs, and also implies that the data distribution encountered in early epochs may exert a significant impact on the final solution.

Existing studies have two key findings: first, perturbations injected in the early training stage cause permanent damage to the final accuracy of the model (Achille et al., 2017; Kleinman et al., 2023), yet no systematic methods have been proposed to take advantage of this observation for performance

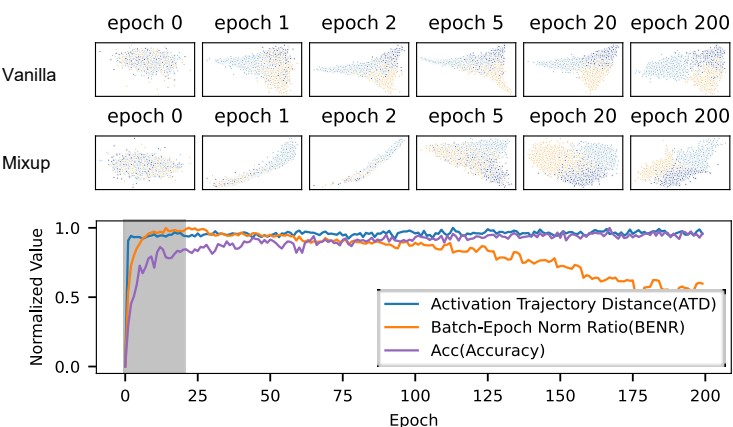

Figure 1: **2D Embedding Visualization** of three selected classes from CIFAR-10 using ResNet18. Comparing two training strategies: (1) vanilla training (2) Training with Input Mixup.

improvement; second, some studies (Kalra & Barkeshli, 2023; Geiger et al., 2018; Tan et al., 2024) have explored early training dynamics based on phase transition theory in physics. If this theory applies to the early training stage, an **order parameter** should exist to characterize the order degree of the model state. At the initial training stage, the model lacks effective classification capability—for most classification tasks (e.g., CIFAR-100), its accuracy is close to zero, corresponding to a highly disordered phase. As training progresses to the middle and late stages, various metrics stabilize, accuracy increases steadily, and the system gradually enters a more ordered phase. Thus, **we propose that accuracy (or loss function value) can be regarded as a macroscopic indicator analogous to a quasi-order parameter, which characterizes the system's order degree.**

Although numerous studies have explored the key characteristics of the early training phase, to the best of our knowledge, no existing work has directly proposed practical optimization methods based on these findings; more detailed information on relevant studies is available in the Appendix A.

Based on this perspective, this study defines the phase corresponding to the light gray region in Figure1 as the model's *Enlightenment Period*: a critical transition where the model shifts from a disordered to an ordered phase. Specifically introduced for neural network training, this period overlaps with the stage of most intense dynamics and is confined to intense parameter fluctuations. Experimental results show that it spans from the beginning of training until the accuracy reaches approximately 50%, which uses a different definition criterion from the 'early training stage' defined in other studies. Through theoretical derivation and experimental verification, we confirm that during the Enlightenment Period, the distribution of training samples has a significant impact on training performance. Using the dynamic characteristics of this phase effectively can ultimately improve the overall performance of the model.

**The main contributions of this paper are as follows:**

**(a)** We identify a brief but critical 'Enlightenment Period' at the start of deep neural network training, where representations transition from disorder to order. During this phase, Mixup induces a **Gradient Interference Effect** that disrupts boundary refinement; however, this interference weakens with larger data or model size, a phenomenon we term the **Gradient Interference Diminishing Effect**. At the same time, Mixup also provides an **Activation Revival Effect** that restores saturated neurons.

**(b)** Building on these insights, we propose three strategies tailored to different scenarios: the **Mixup Pause Strategy** for small-scale settings to mitigate interference, the **Alpha Boost Strategy** for large-scale underfitting models(the training sample set size is large, the model parameter size is large) to enhance activation revival, and the **High-Loss Removal Strategy** for domains where Mixup is inapplicable (e.g., time-series, LLMs).

**(c)** Extensive experiments across diverse models (ResNet, ViT) and datasets (CIFAR, ImageNet) demonstrate that these strategies yield statistically significant performance gains, offering a new perspective on improving DNN training by exploiting early-phase dynamics.

## 2 ANALYSIS

Our analysis begins with a modeling setup grounded in classification accuracy to capture the essential dynamics of the Enlightenment Period. Section 2.2 examines how Mixup introduces disruptive gradient interference, Sections 2.3–2.4 show how this interference diminishes and how activation revival emerges as a beneficial counter-effect.

### 2.1 MODELING SETUP

We study binary classification with input-label pairs $(\boldsymbol{x}, y) \sim P(\boldsymbol{x}, y)$, where $\boldsymbol{x} \in \mathbb{R}^d$ and $y \in \{0, 1\}$. The model produces the prediction probability $\hat{y} = \sigma(\boldsymbol{\theta}^\top \boldsymbol{x})$ using a linear score function $f(\boldsymbol{x}; \boldsymbol{\theta}) = \boldsymbol{\theta}^\top \boldsymbol{x}$ and the sigmoid activation $\sigma(z) = \frac{1}{1+e^{-z}}$. The training objective is the binary cross-entropy loss $L(y, f) = -y \log \sigma(f) - (1 - y) \log(1 - \sigma(f))$.

For Mixup with a ratio $\lambda \in [0, 1]$, two samples $(\boldsymbol{x}_i, y_i)$ and $(\boldsymbol{x}_j, y_j)$ are interpolated to form a new sample: $\boldsymbol{x}_{\text{mix}} = \lambda \boldsymbol{x}_i + (1 - \lambda) \boldsymbol{x}_j$ and $y_{\text{mix}} = \lambda y_i + (1 - \lambda) y_j$, $\lambda \sim \text{Beta}(\alpha, \alpha)$, where $\alpha > 0$ is a hyperparameter controlling the strength of the interpolation. In iteration $t$, the parameters are updated via gradient descent in this mixed sample: $\boldsymbol{\theta}_{t+1} = \boldsymbol{\theta}_t - \eta \nabla_{\boldsymbol{\theta}} L(y_{\text{mix}}, f(\boldsymbol{x}_{\text{mix}}; \boldsymbol{\theta}_t))$. A detailed justification for this modeling approach for analyzing Mixup during the Enlightenment Period is provided in the AppendixC.

### 2.2 ENLIGHTENMENT PERIOD MIXUP GRADIENT INTERFERENCE

During the Enlightenment Period, the model frequently misclassifies samples: positive samples tend to yield $f(\boldsymbol{x}_i; \boldsymbol{\theta}) = \boldsymbol{\theta}^T \boldsymbol{x}_i < 0$, and negative samples yield $f(\boldsymbol{x}_j; \boldsymbol{\theta}) = \boldsymbol{\theta}^T \boldsymbol{x}_j > 0$. Under the assumption that the non-linearity of the activation dominates, we approximate $\sigma(f_i) \approx 0$, where $f_i = \boldsymbol{\theta}^T \boldsymbol{x}_i$. The gradient of the loss for the positive sample is $\nabla_{\boldsymbol{\theta}} L_i \approx -\boldsymbol{x}_i$, and similarly, for the negative sample, it is $\nabla_{\boldsymbol{\theta}} L_j \approx \boldsymbol{x}_j$. Thus, the total gradient from standard training (without Mixup) is approximately

$$\nabla_{\text{vanilla}} = \nabla L_i + \nabla L_j \approx -\boldsymbol{x}_i + \boldsymbol{x}_j \qquad (1)$$

This update direction is effective for correcting misclassifications.

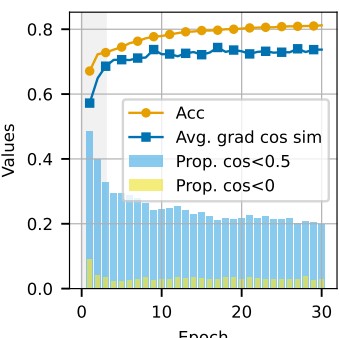

Figure 2: Statistics of Cosine Similarity Between Vanilla and Mixup Gradient Updates. **Avg. grad cos sim**: Average cosine similarity of gradients across all sample pairs; **Prop. (cos < 0.5)**: Proportion of sample pairs with a gradient angle exceeding $60°$; **Prop. (cos < 0)**: Proportion of sample pairs with a gradient angle exceeding $90°$

In the special case of $\lambda = 0.5$, when interpolating a positive sample $(\boldsymbol{x}_i, y_i = 1)$ and a negative sample $(\boldsymbol{x}_j, y_j = 0)$, we obtain $\boldsymbol{x}_{\text{mix}} = \frac{1}{2}(\boldsymbol{x}_i + \boldsymbol{x}_j)$ and $y_{\text{mix}} = \frac{1}{2}$. The loss is $L_{\text{mix}} = -0.5 \log \sigma(f_{\text{mix}}) - 0.5 \log(1 - \sigma(f_{\text{mix}}))$, where $f_{\text{mix}} = \boldsymbol{\theta}^T \boldsymbol{x}_{\text{mix}} = \frac{1}{2}(f_i + f_j)$. Given $|f_i| > |f_j|$ and $f_i < 0$ in early training, it follows that $f_{\text{mix}} < 0$, and therefore $\sigma(f_{\text{mix}}) \approx 0$. The resulting gradient is as follows:

$$\nabla_{\text{mix}} \approx -\frac{1}{4}(\boldsymbol{x}_i + \boldsymbol{x}_j) \qquad (2)$$

This Mixup-induced gradient introduces a deviation from the original update direction. To simulate the scenario of varying $\lambda$ in real Mixup training, this gradient summation (of original and Mixup samples) is adopted to approximate the actual total gradient:

$$\nabla_{\text{total}} = \nabla_{\text{vanilla}} + \nabla_{\text{mix}} \approx -\frac{5}{4}\boldsymbol{x}_i + \frac{3}{4}\boldsymbol{x}_j \qquad (3)$$

Under the common assumptions that input samples are normalized and high-dimensional, $\boldsymbol{x}_i$ and $\boldsymbol{x}_j$ can be considered to have approximately equal norms and be nearly orthogonal. In such cases, the vector $\boldsymbol{x}_i + \boldsymbol{x}_j$ is approximately orthogonal to $\boldsymbol{x}_j - \boldsymbol{x}_i$, meaning the Mixup gradient contributes primarily in a direction orthogonal to the useful update provided by $\nabla_{\text{vanilla}}$. As a result, Mixup will initially interfere with efficient boundary refinement during the early training stages.

The above reasoning is verified in Figure 2. For this experiment, a 2-layer MLP with a hidden layer of size 256 and sigmoid activation was trained on two-class CIFAR-10. Before each training epoch, one sample was randomly selected from each of the two classes, and the following steps were conducted without actual parameter updates: (1) calculating the vanilla gradients of the two samples; (2) performing 50% weight mixing on the two samples and computing the gradients of the mixed samples (Mixup gradients). After iterating through the entire dataset, the cosine similarity between vanilla and Mixup gradients was analyzed. During early training (first few epochs), the average angle between Mixup and vanilla gradients was very large. With increasing accuracy, the angle between the two gradients diminished quickly, and the instances of large angles were sharply reduced. This demonstrates that Mixup gradients indeed deviate from the main gradient direction in early training, introducing interfering gradient components.

In the later training stages, most samples are correctly classified, meaning the model satisfies $f(\boldsymbol{x}_i; \boldsymbol{\theta}) = \boldsymbol{\theta}^\top \boldsymbol{x}_i \gg 0$ for a positive sample $\boldsymbol{x}_i$ and $f(\boldsymbol{x}_j; \boldsymbol{\theta}) = \boldsymbol{\theta}^\top \boldsymbol{x}_j \ll 0$ for a negative one. For a Mixup sample (assuming $\lambda = 0.5$), the output is $f_{\text{mix}} = f(\boldsymbol{x}_{\text{mix}}; \boldsymbol{\theta}) = \frac{1}{2}\big(f(\boldsymbol{x}_i; \boldsymbol{\theta}) + f(\boldsymbol{x}_j; \boldsymbol{\theta})\big)$. If the model has not yet converged to the optimal decision boundary (i.e., the midpoint between positive and negative samples), then $f_{\text{mix}} \neq 0$. Consequently, the gradient $\nabla_{\boldsymbol{\theta}} L_{\text{mix}}$ is also non-zero, which still drives effective parameter updates.

## 2.3 GRADIENT INTERFERENCE DIMINISHING EFFECT

Theoretically, the negative interference mentioned above is not present in all scenarios. The following formulas prove that its effect diminishes as the training data size or the model parameter size grows.

### 2.3.1 GRADIENT INTERFERENCE DIMINISHING EFFECT WITH SAMPLE SIZE

We consider training samples drawn from two distributions: $P_+(\boldsymbol{x})$ for the positive class ($y = 1$) and $P_-(\boldsymbol{x})$ for the negative class ($y = 0$), with respective expectations $\mu_+ = \mathbb{E}_{P_+}[\boldsymbol{x}]$ and $\mu_- = \mathbb{E}_{P_-}[\boldsymbol{x}]$. Given a sufficiently large sample size $N$, we define the distributional difference as $\Delta\mu = \mu_- - \mu_+$ and analyze how Mixup affects the gradient dynamics as $N$ increases.

In standard training, as the sample size $N \to \infty$, the total gradient satisfies $\nabla_{\text{vanilla}} = \sum_{k=1}^{N} \nabla L_i^{(k)} + \sum_{k=1}^{N} \nabla L_j^{(k)} \approx -\sum_{k=1}^{N} \boldsymbol{x}_i^{(k)} + \sum_{k=1}^{N} \boldsymbol{x}_j^{(k)}$. By the Law of Large Numbers, we have $\nabla_{\text{vanilla}} \xrightarrow{P} N \cdot \Delta\mu$, which implies $\|\nabla_{\text{vanilla}}\|_2 = O(N)$. This establishes that the useful gradient signal scales linearly with sample size.

For **Mixup gradient perturbations** which represent the deviation from the standard gradient direction, we define $\boldsymbol{\delta}^{(k)}$ as the difference between the gradient (Eq.equation 3) in Mixup training and the vanilla gradient (Eq. equation 1). This gives:

$$\boldsymbol{\delta}^{(k)} = \nabla L_{\text{mix}}^{(k)} \approx \pm \frac{1}{4}(\boldsymbol{x}_i^{(k)} + \boldsymbol{x}_j^{(k)}). \tag{4}$$

Since the sample pair $(\boldsymbol{x}_i^{(k)}, \boldsymbol{x}_j^{(k)})$ is drawn randomly and independently from the training set at each step, these perturbations can be modeled as independent and identically distributed (i.i.d.) random variables. Consequently, $\mathbb{E}[\boldsymbol{\delta}^{(k)}] \approx 0$, which means Mixup gradient perturbations do not introduce systematic bias in the gradient direction. We also model these perturbations as isotropic, i.e., their covariance matrix is $\text{Cov}(\delta^{(k)}) = \sigma^2 I$.

Let $\nabla_{\text{mix}} = \sum_{k=1}^{N} \boldsymbol{\delta}^{(k)}$ denote the Mixup gradient perturbations of total training set. Since each $\delta^{(k)}$ is independent and identically distributed with zero mean, the Central Limit Theorem yields $\|\nabla_{\text{mix}}\|_2 = O_p(\sqrt{N})$. More precisely, defining $\Sigma_{\text{mix}} = \text{Var}(\boldsymbol{\delta}^{(k)})$, we obtain $\mathbb{E}\left[\|\nabla_{\text{mix}}\|_2\right] \approx \sqrt{N \cdot \text{tr}(\Sigma_{\text{mix}})}$.

We define the **Gradient Interference Strength** as $\epsilon_N = \frac{\|\nabla_{\text{mix}}\|_2}{\|\nabla_{\text{vanilla}}\|_2}$. Combining the results above, we have $\epsilon_N = O_p\left(\frac{1}{\sqrt{N}}\right)$. Consequently, $\lim_{N \to \infty} \epsilon_N = 0$ in probability, demonstrating that Mixup interference effects asymptotically vanish as the sample size increases.

### 2.3.2 GRADIENT INTERFERENCE DIMINISHING EFFECT WITH PARAMETER SIZE

Consider an infinitely wide neural network with parameter size $D$ (parameters $\boldsymbol{\theta} \in \mathbb{R}^D$). Let $\boldsymbol{g} = \nabla_{\boldsymbol{\theta}} L(\boldsymbol{\theta})$ denotes the vanilla gradient, and $\boldsymbol{\delta}$ denotes Mixup gradient perturbations in Eq.equation 4. The parameter update direction is $\Delta \boldsymbol{\theta} = -\eta(\boldsymbol{g} + \boldsymbol{\delta})$.

Using a first-order Taylor expansion of $L(\boldsymbol{\theta} + \Delta \boldsymbol{\theta})$, we get

$$L(\boldsymbol{\theta} + \Delta \boldsymbol{\theta}) \approx L(\boldsymbol{\theta}) + \boldsymbol{g}^\top \Delta \boldsymbol{\theta} = L(\boldsymbol{\theta}) - \eta \boldsymbol{g}^\top (\boldsymbol{g} + \boldsymbol{\delta}) \tag{5}$$

Thus, the loss reduction is

$$\Delta L = L(\boldsymbol{\theta}) - L(\boldsymbol{\theta} + \Delta \boldsymbol{\theta}) \approx \eta \|\boldsymbol{g}\|_2^2 + \underbrace{\eta \boldsymbol{g}^\top \boldsymbol{\delta}}_{\Delta L_{\text{noise}}} \tag{6}$$

where $\eta \|\boldsymbol{g}\|_2^2$ is the deterministic descent term (driven by the vanilla gradient) and $\Delta L_{\text{noise}} = \eta \boldsymbol{g}^\top \boldsymbol{\delta}$ is the perturbation-induced stochastic term (caused by Mixup gradient perturbations).

**Variance analysis**: To quantify the stochastic term's variance, we compute

$$\text{Var}(\Delta L_{\text{noise}}) = \mathbb{E}[(\boldsymbol{g}^\top \boldsymbol{\delta})^2] - (\mathbb{E}[\boldsymbol{g}^\top \boldsymbol{\delta}])^2 \tag{7}$$

Since $\mathbb{E}[\boldsymbol{\delta}] = \boldsymbol{0}$, the second term vanishes. By the cyclic property of the Trace Operator,

$$\mathbb{E}[(\boldsymbol{g}^\top \boldsymbol{\delta})^2] = \mathbb{E}[\boldsymbol{\delta}^\top \boldsymbol{g} \boldsymbol{g}^\top \boldsymbol{\delta}] = \text{Tr}(\boldsymbol{g} \boldsymbol{g}^\top \mathbb{E}[\boldsymbol{\delta} \boldsymbol{\delta}^\top]) = \sigma^2 \text{Tr}(\boldsymbol{g} \boldsymbol{g}^\top) = \sigma^2 \|\boldsymbol{g}\|_2^2 \tag{8}$$

Thus, $\text{Var}(\Delta L_{\text{noise}}) = \eta^2 \sigma^2 \|\boldsymbol{g}\|_2^2$. The relative fluctuation intensity (ratio of fluctuation magnitude to expected loss reduction) is then

$$\frac{\text{std}(\Delta L_{\text{noise}})}{\mathbb{E}[\Delta L]} \approx \frac{\sqrt{\eta^2 \sigma^2 \|\boldsymbol{g}\|_2^2}}{\eta \|\boldsymbol{g}\|_2} = \frac{\sigma}{\|\boldsymbol{g}\|_2} \tag{9}$$

**Asymptotic behavior with parameter size**: In large-scale networks, the gradient norm squared scales with $D$, i.e., $\|\boldsymbol{g}\|_2^2 \sim D\bar{g}^2$ ($\bar{g}$ is the average per-parameter gradient magnitude). Substituting $\|\boldsymbol{g}\|_2 \sim \sqrt{D}\bar{g}$ into the relative fluctuation gives $\frac{\sigma}{\|\boldsymbol{g}\|_2} \sim \frac{\sigma}{\sqrt{D}\bar{g}} = O_p(1/\sqrt{D})$, which tends to 0 as $D \to \infty$, meaning $\Delta L_{\text{noise}}$ has an increasingly negligible impact on $\Delta L$ as parameter size grows.

Therefore, as parameter size $D$ increases, the impact of Mixup perturbations (reflected by $\Delta L_{\text{noise}}$) diminishes. Consequently, large models naturally average out Mixup gradient perturbations, leading to more stable training compared to small models.

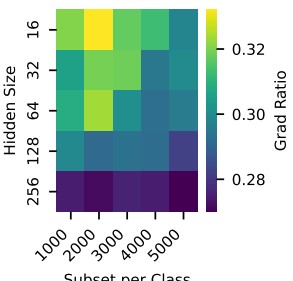

### 2.3.3 EXPERIMENTAL VALIDATION

Although theory predicts $O_p\left(\frac{1}{\sqrt{x}}\right)$, in practice we only expect to observe an approximate trend due to the complexity of the model.

To verify that the effect diminishes: experiments were conducted using two classes of data from the CIFAR-10 dataset and a two-layer MLP with sigmoid activation. During the first epoch, before any parameter updates, the model gradients based on the entire training dataset were calculated separately

Figure 3: Experimental results for verifying gradient interference and its diminishing effect

under the vanilla training strategy and the Mixup strategy. A ratio $r$ was defined to quantify gradient interference: $r = \frac{\|\nabla_{\text{Mixup}}^\perp\|_2}{\|\nabla_{\text{vanilla}}\|_2}$. Here, $\|\nabla_{\text{Mixup}}^\perp\|_2$ denotes the L2 norm of the component of the Mixup gradient ($\nabla_{\text{Mixup}}$) that is perpendicular to the vanilla gradient ($\nabla_{\text{vanilla}}$), and $\|\nabla_{\text{vanilla}}\|_2$ denotes the L2 norm of the vanilla gradient. **Grad Rate** refers to the value of $r$ in the first epoch. In Figure 3, when the sample size or the size of the model parameter (e.g. the number of neurons in the hidden layer) increases, the interference weakens; this confirms the Gradient Interference Diminishing Effect.

### 2.4 ACTIVATION REVIVAL EFFECT

During the Enlightenment Period, violent and disorderly fluctuations in model parameters can cause activation saturation and vanishing gradients. Mixup can reverse this issue by restoring gradient updates for these neurons. For example, if the input value of a ReLU activation function is much less than 0 (a key characteristic of ReLU), it does not update the gradients. When mixed with a sample whose input value is greater than 0, their weighted average may return to the normal range, thus restoring gradient updates, which is the manifestation of the Activation Revival Effect.

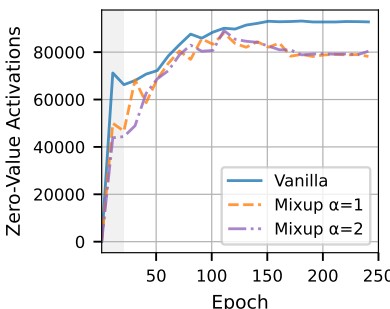

Figure 4: Zero-Value Activations per Sample (ResNet18 on CIFAR-100: Vanilla vs Mixup)

As shown in Figure 4, during the training of ResNet18 on the CIFAR-100 dataset, the average number of zero activation values per sample from the validation set is presented for the vanilla and Mixup strategies after each epoch. Key observations are as follows: under the vanilla strategy, the average number of zero activation values per sample shows a basically monotonic increase. In contrast, that of the Mixup strategy peaks in the middle stage of training and then decreases. Thus, we can conclude that Mixup can restore gradients for neurons with activation saturation. Additionally, during the Enlightenment Period (light gray area), the average number of zero activation values of the Mixup strategy is significantly lower than that of the vanilla strategy. Furthermore, the higher the alpha value of the Mixup, the more pronounced this effect becomes. This allows neurons to function fully during this period, which is particularly critical for improving the performance of underfitting networks, as they require more effective parameters to fit samples.

## 3 MIXUP PAUSE AND ALPHA BOOST

The analyses in the preceding sections reveal a fundamental trade-off in applying Mixup during the Enlightenment Period. On one hand, the Mixup Gradient Interference Effect can introduce disruptive gradient components that hinder the stable formation of early representations. On the other hand, the Activation Revival Effect offers a crucial optimization benefit by mitigating neuron saturation and rescuing vanishing gradients.

The optimal approach depends on which of these two effects is dominant, a condition largely determined by the training scale.

Based on this trade-off, we propose two scenario-driven strategies for the Enlightenment Period:

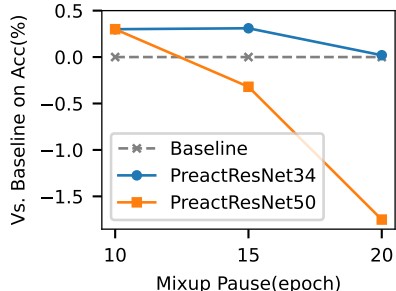

Figure 5: Acc Improvement of Models Trained with the Mixup Pause Strategy Over the Baseline on Cifar100

**Mixup Pause Strategy**: For small-scale scenarios, we propose disabling Mixup during the Enlightenment Period.

**Alpha Boost Strategy**: For large-scale scenarios with underfitting, we propose increasing Mixup's $\alpha$ value (compared to the baseline $\alpha$ used in full-cycle training) during the Enlightenment Period.

As illustrated in Figure 5, pausing Mixup yields the most consistent and prominent performance gains for smaller models. This observation provides empirical support for the Gradient Interference Diminishing Effect with Increasing Parameter Size. The results presented in Table 1 further reinforce this trend: for the same model, the Mixup Pause Strategy delivers significant benefits when applied to smaller data subsets, specifically the 10%-70% subsets of ImageNet-1K. However, this advantage fades when the full ImageNet-1K dataset is used, which has a larger sample size. At this stage, the Alpha Boost Strategy starts to outperform the baseline. This phenomenon is consistent with the Gradient Interference Diminishing Effect with Increasing Sample Size.

Table 1: Performance/Data Scale (ViT-T on ImageNet-1k)

| Strategy | 10% | 30% | 70% | 100% |
|---|---|---|---|---|
| Vanilla | 40.41 | 59.46 | 67.6 | 73.9 |
| Mixup Pause | **41.10** | **59.82** | **67.9** | 73.5 |
| Alpha Boost | 40.01 | 59.05 | 67.5 | **74.3** |

Table 2: Acc at Optimal Enlightenment Period End (Various Models and Datasets)

| Model | Dataset | Acc |
|---|---|---|
| ResNet18 | FOOD101 | 66 |
| ResNet34 | CIFAR-100 | 43 |
| PreActResNet34 | CIFAR-100 | 52 |
| PreActResNet50 | Tiny-ImageNet | 42 |
| ViT-T | ImageNet-1K | 40 |

In summary, while it is theoretically infeasible to define a precise quantitative boundary that distinguishes small-scale scenarios from large-scale scenarios with underfitting, experimental evidence offers a practical reference. This boundary roughly matches the scale of ViT-T when trained on the 100% ImageNet-1K dataset. Specifically, scenarios smaller than this scale are well-suited for the Mixup Pause Strategy. For scenarios that are at or larger than this scale and also exhibit underfitting, the Alpha Boost Strategy provides greater performance benefits.

When considering the Gradient Interference Diminishing Effect with increasing sample size, the impact of batch size must be considered, as its selection is tied to training optimization strategies, model structure, and variations in sample distribution. Although there is extensive research in this area, no widely accepted definitive conclusions have been established to date (Umeda & Iiduka, 2025; Kamo & Iiduka, 2025), so the determination of the batch size in training remains largely empirical.

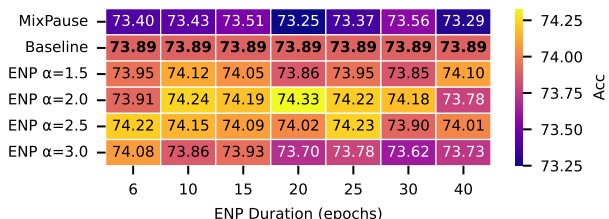

Figure 6: Mean Accuracy of ViT-T on ImageNet-1K vs. Parameters, where the bold text denotes the baseline.

Experiments in Table 1 indicate that once the optimal batch size is identified through experiments, this effect correlates directly with dataset size.

How to determine the duration of the Enlightenment Period? The theoretical model of the Enlightenment Period is based on the early-stage sharp rise of model accuracy from low to high, as shown in Subsection 2.2. Meanwhile, Table 2 shows that the optimal Enlightenment Period typically ends around the point when the model accuracy reaches 50%. Thus, testing around 50% Acc will help identify the optimal Enlightenment Period.

# 4 EXPERIMENTS

Our primary experiments validate the Mixup Pause and Alpha Boost strategies in a diverse set of computer vision models (ResNetShafiq & Gu (2022), PreActResNet(He et al., 2016b;a), ViT(Dosovitskiy et al., 2021)) and datasets (CIFAR-10/100, Tiny-ImageNet(Krizhevsky, 2009), ImageNet-1K(Deng et al., 2009)), totaling 16 different sets of hyperparameters. The key results are presented in the main text, with full details and additional results available in the AppendixD. For tasks where Mixup is inapplicable, such as time-series forecasting and LLM text generation, we evaluate our proposed High-Loss Removal Strategy. The experiments are conducted on NVIDIA H100 GPUs.

For ViT-T/ImageNet-1K in Table3 and Figure6, the values represent the averages of three experiments. For other experiments, we apply one-tailed t-tests to assess statistical significance (see Table3 and Table7).

## 4.1 MIXUP PAUSE AND ALPHA BOOST

As shown in Table 3, Table 7 and Figure 6, we consistently observe across multiple models and datasets that either disabling Mixup in small-scale settings (including fine-tuning) or increasing the

Table 3: Main Results and ablation study for Mixup Pause and Alpha Boost strategies, including training from scratch and fine-tuning. **ENP** denotes the Enlightenment Period.

| Model/Dataset | Mixup $\alpha$ | ENP $\alpha$ | ENP Dur. | Top-1 Acc. (%) Ours | Top-1 Acc. (%) Baseline | Top-1 Acc. (%) $\Delta$ | p-value |
|---|---|---|---|---|---|---|---|
| *Mixup Pause Strategy* | | | | | | | |
| PreactResNet50/Tiny-ImageNet | 2.0 | – | 10 E | 66.37 | 65.92 | **+0.45** | 0.0018 |
| ViT-S/Tiny-ImageNet | 1.0 | – | 10 E | 46.85 | 46.04 | **+0.81** | 0.0128 |
| | 2.0 | – | 10 E | 48.66 | 47.09 | **+1.57** | 0.0003 |
| | 2.0 | – | 25 E† | 46.40 | 47.09 | -0.69 | – |
| ResNet18/CIFAR-10* | 2.0 | – | 10 E | 95.61 | 95.51 | **+0.10** | 0.0310 |
| ResNet34/CIFAR-100* | 2.0 | – | 3 E | 81.76 | 81.31 | **+0.45** | 0.0368 |
| | 2.0 | – | 50 E† | 81.09 | 81.31 | -0.22 | – |
| *Alpha Boost Strategy* | | | | | | | |
| ViT-T/ImageNet-1K | 0.8 | 2 | 20 E | 74.3±0.16 | 73.9±0.20‡ | **+0.4** | – |

* Fine-tuning (employ pretrained models from official PyTorch repository, unfreeze the first convolutional layer and the final hidden layer).
† Denotes an ablation study on the duration of the Mixup Pause (see more in Table7).
‡ Baseline from Liu et al. (2023)
E Epochs

Mixup alpha in large-scale settings during the first few epochs leads to statistically significant performance gains(one-tailed $p < 0.05$) compared to using a single fixed mixup $\alpha$ throughout training. Meanwhile, the ablation experiments in Table 3, Table 7 and Figure 6 show that extending the Enlightenment Period can degrade performance, indicating that the two effects associated with Mixup occur during the Enlightenment Period and are extremely transient.

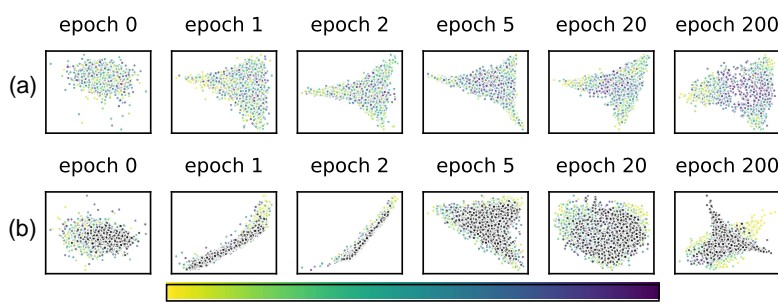

Figure 7: **2D embedding visualization** of three selected classes from the CIFAR-10 dataset using ResNet18. Different colors represent different loss values: (a) vanilla training, (b) Mixup training (black dots denote samples generated with $\lambda = 0.5$). Both the high-loss points in (a) and mixed points in (b) lie on classification boundaries and have similar classification features.

## 4.2 HIGH-LOSS REMOVAL

We observe that high-loss samples share boundary-ambiguous characteristics with Mixup samples, as shown in Figure 7: initially chaotic, high-loss samples form clusters later than low-loss counterparts. Specifically, the formal equivalence between high-loss samples and Mixup samples (i.e., any high-loss sample can be represented as the Mixup of two zero-loss positive/negative samples) is derived in the Appendix B. Thus, similar to the Mixup Pause Strategy, we propose removing a small proportion of the highest-loss samples during the Enlightenment Period. The steps are: first, train a vanilla model on the full training set and record each sample loss; second, select the top k% 'easy'

samples (where k% is a hyperparameter, typically over 85%, see ablation 8); finally, retrain the same model from scratch, using only these 'easy' samples during the Enlightenment Period. Table 4 and Table 5 8 present the results of the High-Loss Removal Strategy. In particular, due to the unique dynamic behavior of large language models Nicolini et al. (2024), their Enlightenment Period may differ from that of conventional models.

Table 4: Performance of the High-Loss Removal Strategy on time series long-term forecasting tasks (look-back=96, prediction length=96). Each new model exhibits consistent MSE reduction of approximately **0.009** over its predecessor.**only** removing a portion of high-loss samples in the first epoch without modifying other parameters enables an additional **0.002** MSE reduction, equivalent to a 22% MSE decrease.

| Model / Dataset | MSE (vs. Baseline) | p-value[†] | Runs (ours/base) |
|---|---|---|---|
| TimeMixer(Wang et al., a) / ECL | 0.154 (vs. 0.156) | 5.41E-06 | 6 / 4 |
| iTransformer(Liu et al.) / ECL | 0.146 (vs. 0.148) | 0.00012 | 4 / 4 |
| TimeXer(Wang et al., b) / ECL | 0.139 (vs. 0.141) | 0.00031 | 4 / 4 |
| iTransformer / Weather | 0.173 (vs. 0.175) | 1.54E-05 | 38 / 14 |
| TimeMixer / Weather | 0.162 (vs. 0.163) | 0.00506 | 21 / 10 |

[*] use top easy 95% samples during the first half of the first epoch on 10 epochs training.

Table 5: Performance of the High-Loss Removal Strategy on a language model.

| Experiment | ENP Duration | Test Loss (vs. Baseline) | Variance | Runs |
|---|---|---|---|---|
| *Model: MiniMind (25M)(Gong, 2024a) / Dataset: pretrain-hq (1.6GB)(Gong, 2024b)* | | | | |
| Main Result | 0-th epoch | 1.9138 (vs. 1.9150) | 1.22E-05 | 30 |
| Ablation Studies | 3-rd epoch | 1.9148 (vs. 1.9150) | 2.34E-06 | 3 |
| | 0-th to 2-nd epochs | 1.9144 (vs. 1.9150) | 1.40E-05 | 8 |
| | 0-th to 3-rd epochs | 1.9187 (vs. 1.9150) | 3.10E-06 | 3 |

[*] use top easy 97% samples. The model was trained for a total of 6 epochs.

# 5 CONCLUSION

This paper focuses on the *Enlightenment Period*, a critical early phase in DNN training, using a three-step framework: (1) establishing a mathematical model for this period via physical phase transition theory; (2) verifying model-derived conclusions (Mixup's Gradient Interference, Gradient Interference Diminishing Effect and Activation Revival Effect) through theoretical derivation and DNN experiments; (3) validating three model-based optimization strategies (Mixup Pause, Alpha Boost, High-Loss Removal) on large-scale models/datasets, with statistically significant performance gains over baselines. In particular, the strategies are scenario-specific: Alpha Boost suits pre-training on large-scale datasets with underfitting like full ImageNet-1K; Mixup Pause and High-Loss Removal excel in small-scale settings, such as data-limited scenarios, time-series tasks, and fine-tuning. This work illuminates the understudied dynamics of early DNN training and provides phase-aware optimization tools that integrate seamlessly with existing pipelines to boost model performance.

## REPRODUCIBILITY STATEMENT

We have made extensive efforts to ensure the reproducibility of our results. The source code and our implementations are available at https://anonymous.4open.science/r/code-A5F1/. Theoretical results are supported by complete proofs and clearly stated assumptions in the Appendix.

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

# A    RELATED WORK

## A.1    EARLY PHASE IN DEEP NETWORKS TRAINING

Some studies suggest that during the early phase of neural network training, the learning process exhibits remarkable simplicity, and its dynamics are effectively approximated by linear models (Hu et al., 2020; Kalimeris et al., 2019). Other research shows that the early phase arises from complex and unstable transient dynamics, which are decisive for the final performance of the trained system and its learned representations (Kleinman et al., 2023; Shen et al., 2024), and are fundamentally different from late stage behavior (Iyer et al., 2023; Leclerc & Madry, 2020; Paul et al., 2021). Numerous studies have demonstrated the special nature of this phase. Table 6 summarizes the lengths and characteristics of the early phase reported in several existing works. Although many studies have explored and derived the key characteristics of the early phase, to the best of our knowledge, no work has directly proposed practical optimization methods based on these findings. In particular, the proposed Enlightenment Period uses a different definition criterion from the 'early training stage' defined in other studies, suggesting that its properties remain unexplored.

Table 6: Related works on early phase in deep networks training

| Dataset | Duration of Early Phase | Key Characteristics |
| --- | --- | --- |
| CIFAR | 25-100 epochs | Regularization-sensitive period (Golatkar et al., 2019) |
| CIFAR-10 | 10-20 epochs | EL2N score validity window (Paul et al., 2021) |
| LLM Training | 30-50% duration | Parameter bifurcation period (Nicolini et al., 2024) |
| CIFAR-10 | 40-60 epochs | Interference-sensitive window (Achille et al., 2017) |
| CIFAR-10/MNIST | 10 steps | Phase transition patterns in learning rates, network depth and width (Kalra & Barkeshli, 2023) |
| CIFAR-10 | first 4000 iterations (10 epochs) | Drastic fluctuation in gradient and weight magnitudes (Frankle et al., 2019) |
| CIFAR-10/ImageNet | dozens of epochs | Generalization-determining phase (Leclerc & Madry, 2020) |
| CIFAR-10 | first 60% of training | Large learning rates reduce gradient variance (Jastrzebski et al., 2020) |

The Enlightenment Period defined in this paper differs from the early phases in the aforementioned studies mainly in that: we regard the Enlightenment Period as a process similar to a physical phase transition, with **classification accuracy serving as the order parameter**. This is the key to the improved training performance achieved in this study: ablation experiments demonstrate that the strategy proposed herein is only effective when applied at the very initial stage of model training, and the strategy loses effectiveness outside this interval. On this basis, we have established a mathematical model for the Enlightenment Period and verified the correctness of the model through experiments. Furthermore, by optimizing the training sample distribution based on the principles of this mathematical model for the Enlightenment Period, we ultimately improve the model performance.

## A.2    DYNAMIC MIXUP

Mixup, a highly successful data augmentation technique, has been widely adopted in various domains, with numerous studies dedicated to dynamically adjusting Mixup throughout the training process (Jin et al., 2024). Previous work on dynamic Mixup (Liu et al., 2021; Bunk et al., 2021; Liu et al., 2022; Mai et al., 2021; Carratino et al., 2022)have primarily focused on two directions: optimizing Mixup strategies (e.g., adversarial perturbation, dynamic masking) or refining Mixup deployment across the entire training cycle . In contrast, our research centers on the Enlightenment Period, which refers to a brief initial training phase. This phase spans from the beginning of training until the accuracy reaches approximately 50% and is defined by the model's transition from a disordered state to an ordered state, driven by the evolution of its internal structure (e.g., parameter fluctuations, activation saturation). Our core contribution lies in regulating the timing of Mixup intervention exclusively within this critical period.

Notably, our strategy does not conflict with the aforementioned Mixup optimization approaches; instead, their joint application can further enhance model performance. For example, a previous study(Zou et al., 2023) proposed that Mixup should be stopped after approximately 100 training epochs to improve performance, and this approach specifically targets the late training phase. Our experimental results demonstrate that combining this late-phase Mixup cessation with our Enlightenment Period-based regulation yields optimal performance.

Consistent with our experimental findings, when training ResNet34 on the CIFAR-100 dataset with a Mixup baseline (which achieves 79.274% top-1 accuracy), disabling Mixup in the last 10 epochs increased the accuracy to 80.338%. Furthermore, turning off Mixup in both the first 10 epochs (i.e., the Enlightenment Period, where the goal is to mitigate gradient interference) and the last 10 epochs further boosted the accuracy to 81.08%. This synergy validates that phase-specific Mixup regulation, which places a particular focus on the Enlightenment Period, complements existing late-phase optimization strategies, thereby unlocking enhanced model performance.

### A.3 CURRICULUM LEARNING

Curriculum Learning (CL) is a strategy that mimics the sequencing observed in human learning processes to train machine learning models. The core principle of CL is 'from easy to difficult', (Zhou et al., 2024). A representative approach 'baby steps' (Bengio et al., 2009), implements automated curriculum scheduling by ordering training data based on loss values. Most current CL approaches are designed based on a 'difficulty measurer + training scheduler' framework(Wang et al., 2021). However, existing studies reveal that neither easy- to-hard nor hard-to-easy curricula can improve model performance(Saglietti et al., 2022; Wu et al.).

In our paper, High-Loss Removal Strategy is fundamentally distinct from traditional CL. Traditional CL adheres to an 'easy-to-hard progressive learning paradigm': it guides models to first construct basic representations using easy samples (Srilakshmi & Sarkar, 2024; Liu et al., 2017), then gradually introduces hard samples (high-loss, complex ones) as core learning targets for long-term training path optimization. Its aim is to improve generalization and convergence across various tasks (e.g., CV, NLP, RL), with a flexible early training phase (where 'early' is merely relative to the middle and late phases, without a definitive end point)(Weinshall et al., 2018; Wang et al., 2023; 2021).

In contrast, High-Loss Removal focuses on **interference exclusion during a specific phase transition period**—it exclusively targets the *Enlightenment Period* (from the beginning of training until the accuracy reaches approximately 50%). During this period, high-loss samples are treated as interference sources (which exacerbate the gradient disturbance induced by the distributional characteristics of high-loss samples in the representational space) and temporarily removed. After the Enlightenment Period, these samples are restored with equal weights to guarantee data integrity. The core goal of the strategy is to facilitate the model's 'disordered-to-ordered' phase transition, rather than achieving direct long-term optimization. In essence, High-Loss Removal serves as 'emergency interference control for a critical phase,' whereas CL functions as 'planned path optimization for long-term training.' (Soviany et al., 2021)

# B PROOF: MIXUP EQUIVALENCE OF NON-ZERO LOSS SAMPLES IN 1D FEATURE SPACE

To further validate the gradient interference mechanism, we prove that in the 1D feature space of our binary classification model, any sample with non-zero loss can be equivalently represented as the Mixup interpolation of two zero-loss positive/negative samples. Moreover, the magnitude of the loss is inversely correlated with the deviation of the Mixup ratio $\lambda$ from 50%.

## B.1 NOTATIONS AND PRELIMINARIES

We define two types of *zero-loss samples* ($L = 0$) that are perfectly classified:

**Positive zero-loss sample**: $(x_+, y_+ = 1)$, satisfying $L(y_+, f_+) = 0$. By the loss in , this implies $\sigma(f_+) = 1$, which requires $f_+ = \theta x_+ \to +\infty$.

**Negative zero-loss sample**: $(x_-, y_- = 0)$, satisfying $L(y_-, f_-) = 0$. Similarly, this implies $\sigma(f_-) = 0$, which requires $f_- = \theta x_- \to -\infty$.

Let the sample of interest be $(x_{\text{loss}}, y)$ with $y \in \{0, 1\}$, finite linear score $f_{\text{loss}} = \theta x_{\text{loss}}$, and non-zero loss $L_{\text{loss}} = L(y, f_{\text{loss}}) > 0$.

### B.2 STEP 1: PROVING THE MIXUP EQUIVALENCE OF $x_{\text{LOSS}}$

The core of Mixup is sample interpolation: given $x_+$ and $x_-$, a mixed sample is defined as $x_{\text{mix}} = \lambda x_+ + (1 - \lambda) x_-$ with $\lambda \in [0, 1]$. We aim to show that *there exists a unique $\lambda$ such that $x_{mix} = x_{loss}$*.

1. Linear score of the mixed sample: for $x_{\text{mix}}$, its linear score is calculated as

$$f_{\text{mix}} = \theta x_{\text{mix}} = \lambda \cdot \theta x_+ + (1 - \lambda) \cdot \theta x_- = \lambda f_+ + (1 - \lambda) f_-. \tag{10}$$

2. Solving for $\lambda$ to match $f_{\text{loss}}$: To make $x_{\text{mix}} = x_{\text{loss}}$, their linear scores must be equal (since $x = \theta^{-1} f$ and $\theta \neq 0$). Setting $f_{\text{mix}} = f_{\text{loss}}$ and rearranging Eq. equation 10 gives:

$$\lambda = \frac{f_{\text{loss}} - f_-}{f_+ - f_-}. \tag{11}$$

3. Verification $\lambda \in [0, 1]$: Since $f_+ \to +\infty$ and $f_- \to -\infty$, both the numerator $f_{\text{loss}} - f_- \to +\infty$ and the denominator $f_+ - f_- \to +\infty$. For simplicity, we take symmetric extreme samples $f_+ = M$ and $f_- = -M$ ($M \to +\infty$), substituting into Eq. equation 11:

$$\lambda = \frac{f_{\text{loss}} + M}{2M} = 0.5 + \frac{f_{\text{loss}}}{2M}. \tag{12}$$

As $M \to +\infty$, the term $\frac{f_{\text{loss}}}{2M} \to 0$, so $\lambda \in (0, 1)$. This confirms the existence of a valid Mixup ratio $\lambda$.

Any sample $x_{\text{loss}}$ with non-zero loss can be equivalently represented as the Mixup interpolation of two zero-loss samples $x_+$ and $x_-$.

#### B.2.1 STEP 2: RELATIONSHIP BETWEEN LOSS MAGNITUDE AND $\lambda$

We analyze how $L_{\text{loss}}$ correlates with the deviation of $\lambda$ from 50%, focusing on both positive ($y = 1$) and negative ($y = 0$) samples.

**Case 1: Positive sample ($y = 1$)**    The loss simplifies to:

$$L(1, f_{\text{loss}}) = -\log \sigma(f_{\text{loss}}) = \log(1 + e^{-f_{\text{loss}}}), \tag{13}$$

where $\sigma(f_{\text{loss}}) = 1/(1 + e^{-f_{\text{loss}}})$. Substituting $f_{\text{loss}} = M(2\lambda - 1)$ (from Eq. equation 12) into Eq. equation 13:

$$L(1, f_{\text{loss}}) = \log\left(1 + e^{-M(2\lambda - 1)}\right). \tag{14}$$

- When $\lambda = 0.5$: $2\lambda - 1 = 0 \implies L = \log(2)$ (maximum loss). - When $\lambda > 0.5$: $2\lambda - 1 > 0 \implies -M(2\lambda - 1) \to -\infty \implies L \to 0$. - When $\lambda < 0.5$: $2\lambda - 1 < 0 \implies -M(2\lambda - 1) \to +\infty \implies L \to +\infty$ (extreme misclassification).

**Case 2: Negative sample ($y = 0$)**    The loss simplifies to:

$$L(0, f_{\text{loss}}) = -\log(1 - \sigma(f_{\text{loss}})) = \log(1 + e^{f_{\text{loss}}}), \tag{15}$$

where $1 - \sigma(f_{\text{loss}}) = e^{-f_{\text{loss}}}/(1 + e^{-f_{\text{loss}}})$. Substituting $f_{\text{loss}} = M(2\lambda - 1)$ into Eq. equation 15:

$$L(0, f_{\text{loss}}) = \log\left(1 + e^{M(2\lambda - 1)}\right). \tag{16}$$

- When $\lambda = 0.5$: $2\lambda - 1 = 0 \implies L = \log(2)$ (maximum loss). - When $\lambda < 0.5$: $2\lambda - 1 < 0 \implies M(2\lambda - 1) \to -\infty \implies L \to 0$. - When $\lambda > 0.5$: $2\lambda - 1 > 0 \implies M(2\lambda - 1) \to +\infty \implies L \to +\infty$ (extreme misclassification).

Let $\Delta = |\lambda - 0.5|$ (deviation of $\lambda$ from 50%). $\Delta$ is monotonically decreasing with $L_{\text{loss}}$: higher $L_{\text{loss}}$ corresponds to smaller $\Delta$ (that is, $\lambda$ closer to 50%), while $L_{\text{loss}} \to 0$ implies $\Delta \to 0.5$ (i.e., $\lambda$ approaches 0 or 1).

### B.3 CONCLUSION

In the 1D feature space of the binary classification model: 1. Any sample with non-zero loss is equivalent to the Mixup interpolation of two zero-loss samples ($x_+$: $y = 1, f \to +\infty$; $x_-$: $y = 0, f \to -\infty$), with the Mixup ratio $\lambda$ given by Eq. equation 11. 2. The magnitude of the loss is inversely correlated with the deviation of $\lambda$ from 50%: the higher the loss, the closer $\lambda$ is to 50%.

## C BENR AND ATD

BENR and ATD are introduced because their **sharp fluctuations in early training** confirm the Enlightenment Period. Furthermore, their laws derived from the aforementioned model in Section 2.1 match experimental observations of BENR and ATD, further justifying the use of this model to explore the principles of the Enlightenment Period.

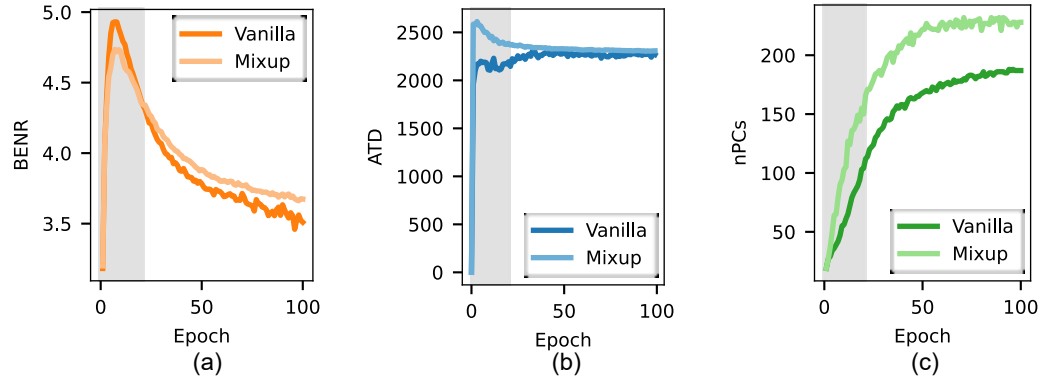

Figure 8: BENR, ATD and changes during the training of PreActResNet-34 on CIFAR-100, covering various training strategies.

### C.1 BATCH-EPOCH NORM RATIO

The Batch-Epoch Norm Ratio (BENR) quantifies the alignment between batch-level and epoch-level parameter updates during neural network optimization. It is defined as the ratio of the sum of L2-norms of batch updates to the L2-norm of epoch updates:

$$\text{BENR}^{(k)} = \frac{\sum \left\| \Delta_{\text{batch}}^{(k)} \right\|_2}{\left\| \Delta_{\text{epoch}}^{(k)} \right\|_2} \tag{17}$$

where $\Delta_{\text{batch}}^{(k)}$ captures the parameter update vector after the $k$-th batch iteration, $\Delta_{\text{epoch}}^{(k)}$ denotes the cumulative parameter update over an entire training epoch, and $\sum$ denotes the summation between batches within an epoch.

The single-batch update ($\Delta_{\text{batch}}$) denotes microscopic parameter fluctuations, analogous to molecular motion in statistical mechanics, with its distribution shaped by SGD perturbations. The cumulative update at the epoch level ($\Delta_{\text{epoch}}$) denotes macroscale parameter responses, analogous to thermodynamic quantities such as pressure and temperature. BENR reflects the relationship between sample sensitivity and optimization trajectory.

Since $\|\Delta\theta_b\|_2 = \eta\|\nabla L_b\|_2$, BENR reflects the cumulative norm of per-batch updates relative to the net parameter change over an epoch.

Assume batch size = 1. In the standard training set, we use two positive samples and one negative sample, trained over three batches. In the Mixup setting, we construct three 50% Mixup samples:

$m_{12} = (x_1 + x_2)/2, m_{13} = (x_1 + x_3)/2, m_{23} = (x_2 + x_3)/2$, assuming that all three samples are misclassified during the Enlightenment Period. For standard training:

$$\nabla L_1 \approx -x_1 \quad \Longrightarrow \quad \|\Delta\theta_1\|_2 = \eta\|x_1\|_2 \tag{18}$$

$$\text{BENR}_{\text{vanilla}} = \frac{\|x_1\|_2 + \|x_2\|_2 + \|x_3\|_2}{\|x_1 + x_2 - x_3\|_2} \tag{19}$$

For Mixup training:

$$\nabla_{\text{total},12} \approx -\frac{3}{2}(\boldsymbol{x}_1 + \boldsymbol{x}_2) \quad \Longrightarrow \quad \|\Delta_{\theta_{12}}\|_2 = \eta\left\|-\frac{3}{2}(\boldsymbol{x}_1 + \boldsymbol{x}_2)\right\|_2 \tag{20}$$

$$\nabla_{\text{total},13} \approx -\frac{5}{4}\boldsymbol{x}_1 + \frac{3}{4}\boldsymbol{x}_3 \quad \Longrightarrow \quad \|\Delta_{\theta_{13}}\|_2 = \eta\left\|-\frac{5}{4}\boldsymbol{x}_1 + \frac{3}{4}\boldsymbol{x}_3\right\|_2 \tag{21}$$

$$\nabla_{\text{total},23} \approx -\frac{5}{4}\boldsymbol{x}_2 + \frac{3}{4}\boldsymbol{x}_3 \quad \Longrightarrow \quad \|\Delta_{\theta_{23}}\|_2 = \eta\left\|-\frac{5}{4}\boldsymbol{x}_2 + \frac{3}{4}\boldsymbol{x}_3\right\|_2 \tag{22}$$

$$\tag{23}$$

$$\nabla_{\text{epoch, mix}} = \nabla_{\text{total},12} + \nabla_{\text{total},13} + \nabla_{\text{total},23} \quad \Longrightarrow \quad \|\Delta_{\theta_{\text{epoch}}}\|_2 \quad = \eta \cdot \|\nabla_{\text{epoch, mix}}\|_2 \tag{24}$$

$$\text{BENR}_{\text{mix}} = \frac{\|\Delta_{\theta_{12}}\|_2 + \|\Delta_{\theta_{13}}\|_2 + \|\Delta_{\theta_{23}}\|_2}{\|\Delta_{\theta_{\text{epoch}}}\|_2} \tag{25}$$

Assuming data are normalized or standardized (that is, $\|x_1\| = \|x_2\| = \|x_3\| = a$) and the vectors $x_1, x_2, x_3$ are orthogonal (a reasonable assumption in high-dimensional settings), then, during the Enlightenment Period:

$$\text{BENR}_{\text{vanilla}} > \text{BENR}_{\text{mix}} \tag{26}$$

This aligns with the observation in Figure 8(a), where BENR is lower for Mixup training in early epochs.

## C.2 ACTIVATION TRAJECTORY DISTANCE

The Activation Trajectory Distance (ATD) quantifies dynamic changes in hidden layer representations during neural network training. Its core mechanism involves:

1. Extracting activation values from the last hidden layer for each validation sample and concatenating them into high-dimensional points.

2. Calculating the L2 geometric distance between these points and the initial state (epoch=0), reflecting the representation shift from the initial to the current state. The ATD is formally defined as follows:

$$\text{ATD}^{(k)} = \left\|\mathbf{A}^{(k)} - \mathbf{A}^{(0)}\right\|_2 \tag{27}$$

where $\mathbf{A}^{(k)}$ denotes the activation vector in the epoch $k$, $\mathbf{A}^{(0)}$ denotes the activation vector in the initialization (epoch = 0), and $\|\cdot\|_2$ denotes the L2 norm.

based on the above derivation of BENR, we obtain:

$$\Delta\theta_{\text{vanilla}} = -\eta(x_1 + x_2 - x_3), \quad \Delta\theta_{\text{mix}} = -\eta \cdot \left(\frac{11}{4}x_1 + \frac{11}{4}x_2 - \frac{3}{2}x_3\right) \tag{28}$$

Given our model $A(x; \theta) = \theta^T x$ and from the BENR derivation $\|\Delta\theta_{\text{vanilla}}\| > \|\Delta\theta_{\text{mix}}\|$. Assuming sufficient diversity in the validation samples $x$, this leads to:

$$\text{Var}(\Delta\text{ATD}_{\text{vanilla}}) > \text{Var}(\Delta\text{ATD}_{\text{mix}}) \tag{29}$$

which is consistent with the observation in Figure 8(b) that Mixup reduces ATD fluctuation during the Enlightenment Period.

Figure8(b) shows that ATD converges immediately and stabilizes with minimal fluctuations after the Enlightenment Period, mainly due to the model's hidden layer activation vectors changing from "disordered exploration" to "constraint by the low-dimensional data manifold" . Figure8(c) presents PCA results of 512-dimensional activation vectors (from the last hidden layer of all training samples) after each epoch.

During the Enlightenment Period, the network lacks stable representations; hidden layer activation vectors fluctuate disorderly in high-dimensional space (even including data-irrelevant directions), driving ATD up. After this period, the network gradually aligns with the low-dimensional manifold of the data. Activation vector variations are confined to the subspace of the manifold (no random divergence), so ATD stops growing and stabilizes.

## D SUPPLEMENTARY EXPERIMENTAL RESULTS

Table7and Table8 are some additional experimental data. We use multiple hyperparameter sets (denoted as HP-n) to validate the universality of the Enlightenment Period across diverse experimental setups (see Table9). The detailed settings for all experiments are provided in the source code package. All training was conducted on NVIDIA H100 and P100 GPUs. Training on ImageNet-1K took approximately 9 hours in a configuration with eight H100 GPUs.

As shown in Table 7, it has been consistently observed in various models and datasets that simply disabling Mixup during the initial few epochs yields statistically significant performance improvements compared to using Mixup throughout all training epochs.

In Table10, for training scenarios using Input MixupZhang et al. (2018) and CutMixYun et al. (2019), Input Mixup is a continuous linear interpolation with continuous changes as the mixing ratio $\lambda$ varies. In contrast, CutMix exhibits discontinuous changes with $\lambda$ due to its block-wise region replacement nature. In this case, if the Mixup Pause strategy is applied, both Mixup methods are disabled during the Enlightenment Period; if the Alpha Boost strategy is adopted, only the alpha value of Input Mixup is increased during the Enlightenment Period, while CutMix's alpha value remains unchanged at the baseline.

## E LIMITATIONS

There remains a broader scope for further research. For example, the High-Loss Removal strategy is theoretically analogous to the Mixup Pause strategy. One promising research direction is to explore whether increasing the proportion of high-loss samples during the Enlightenment Period by following the logic of the Alpha Boost strategy can similarly enhance model performance.

## F USE OF LARGE LANGUAGE MODELS

In preparing this manuscript, we employed a large language model (LLM) solely to aid in polishing the writing. All conceptualization, experimental design, implementation, analysis, and conclusions are our own.

Table 7: Mixup Pause Strategy cross-Model & dataset performance

| Dataset/Model | Method | ENP Duration | Top-1 (%) | Δ | Variance | t-test | Runs |
|---|---|---|---|---|---|---|---|
| PreactResNet34 Cifar100 HP-1 | vanilla (no Mixup) | – | 75.65 | – | 0.0525 | – | 7 |
| | baseline (all Mixup) | – | 79.87 | – | 0.0561 | – | 11 |
| | Ours (a) | 0–6 | 80.04 | +0.17 | 0.0550 | 0.0535 | 12 |
| | | 0–10 | 80.17 | +0.30 | 0.0798 | 0.0065 | 12 |
| | | 0–15 | 80.18 | +0.31 | 0.0397 | 0.0019 | 11 |
| | | 0–20 | 79.89 | +0.02 | 0.2770 | 0.4630 | 6 |
| | Ablation | 0–25 | 76.89 | -2.99 | 5.5230 | – | 3 |
| | | 0–35 | 77.22 | –2.65 | 6.5971 | – | 3 |
| | | 0–45 | 79.22 | –0.65 | 1.1861 | – | 3 |
| | | 0–60 | 78.62 | –1.25 | 3.7357 | – | 5 |
| | | 35–50 | 79.77 | –0.10 | 0.1117 | – | 3 |
| | | 85–100 | 79.86 | –0.01 | 0.0277 | – | 3 |
| | | 285–300 | 79.93 | +0.06 | 0.0097 | – | 3 |
| | | 385-400 | 79.72 | -0.16 | 0.0005 | – | 3 |
| | | 435-450 | 79.63 | -0.24 | 0.1329 | – | 3 |
| PreActResNet50 Cifar100 HP-2 | vanilla (no Mixup) | – | 76.88 | – | 5.9000 | – | 12 |
| | baseline (all Mixup) | – | 81.03 | – | 0.1062 | – | 17 |
| | Ours (a) | 0–10 | 81.32 | +0.30 | 0.1200 | 0.0072 | 17 |
| | | 0–15 | 80.71 | -0.32 | 2.5404 | 0.2138 | 9 |
| | | 0–20 | 79.28 | -1.75 | 11.2345 | 0.0189 | 6 |
| PreactResNet50 Tiny-Imagenet HP-3 | vanilla (no Mixup) | – | 62.37 | – | 0.0880 | – | 3 |
| | baseline (all Mixup) | – | 65.92 | – | 0.0255 | – | 8 |
| | Ours (a) | 0–10 | 66.37 | +0.45 | 0.0935 | 0.0018 | 6 |
| | | 0–15 | 65.29 | -0.63 | 6.8592 | 0.2546 | 6 |
| PreactResNet34 Cifar100 HP-4 | baseline (all Manifold Mixup) | – | 81.575 | – | 0.1079 | – | 13 |
| | Ours (a) | 0–6 | 81.783 | +0.208 | 0.0136 | 0.0175 | 14 |
| | | 0–10 | 81.765 | +0.191 | 0.0365 | 0.0414 | 13 |
| | | 0–15 | 81.723 | +0.148 | 0.0839 | 0.0919 | 20 |
| ResNet18 FOOD101 HP-5 | vanilla (no Mixup) | – | 78.21 | – | 0.0620 | – | 5 |
| | baseline (all Mixup) | – | 81.26 | – | 0.0371 | – | 9 |
| | Ours (a) | 0–1 | 81.45 | +0.20 | 0.0599 | 0.0608 | 5 |
| | | 0–2 | 81.44 | +0.18 | 0.0448 | 0.0632 | 5 |
| | | 0–6 | 81.47 | +0.22 | 0.1476 | 0.0877 | 5 |
| | | 0–10 | 81.48 | +0.23 | 0.0557 | 0.0374 | 5 |
| | | 0–15 | 81.49 | +0.24 | 0.0385 | 0.0117 | 8 |
| | | 0–20 | 81.51 | +0.25 | 0.0415 | 0.0206 | 5 |
| | Ablation | 0–30 | 80.35 | -0.90 | 0.0496 | – | 3 |
| | | 0–50 | 81.09 | -0.16 | 0.0286 | – | 3 |
| | | 15–30 | 81.07 | -0.18 | 0.0269 | – | 4 |
| | | 35–50 | 80.66 | -0.60 | 0.0305 | – | 4 |
| | | 65–80 | 80.48 | -0.77 | 0.0975 | – | 4 |
| | | 95–110 | 81.23 | -0.03 | 0.0011 | – | 4 |
| ResNet34 Cifar100 HP-6 | vanilla (no Mixup) | – | 73.42 | – | 0.0429 | – | 6 |
| | baseline (all Mixup) | – | 78.89 | – | 0.1114 | – | 6 |
| | Ours (a) | 0–1 | 78.96 | +0.06 | 0.0234 | 0.3675 | 4 |
| | | 0–2 | 79.08 | +0.18 | 0.0174 | 0.1668 | 4 |
| | | 0–6 | 79.34 | +0.45 | 0.0310 | 0.0200 | 4 |
| | | 0–10 | 79.26 | +0.37 | 0.0489 | 0.0248 | 6 |
| | | 0–15 | 79.23 | +0.34 | 0.0312 | 0.0269 | 6 |
| | | 0–20 | 79.11 | +0.22 | 0.0326 | 0.0945 | 6 |
| ResNet34 CIFAR-100 HP-7 | baseline (all Mixup) | – | 78.96 | – | 0.0464 | – | 10 |
| | Ours (a) | 0–6 | 79.35 | +0.39 | 0.0568 | 0.0004 | 12 |
| | | 0–10 | 79.21 | +0.25 | 0.0288 | 0.0046 | 10 |
| ResNet34 Cifar100 Paramate8 | vanilla (no Mixup) | – | 73.52 | – | 0.0267 | – | 5 |
| | baseline (all Mixup) | – | 79.00 | – | 0.0289 | – | 5 |
| | Ours (a) | 0–1 | 79.17 | +0.17 | 0.0505 | 0.1023 | 5 |
| | | 0–6 | 79.27 | +0.27 | 0.0362 | 0.0221 | 5 |
| | | 0–10 | 79.44 | +0.44 | 0.1502 | 0.0235 | 5 |
| | | 0–15 | 79.26 | +0.26 | 0.0078 | 0.0087 | 5 |
| | | 0–20 | 79.16 | +0.16 | 0.0194 | 0.0690 | 5 |
| | | 0–25 | 79.25 | +0.25 | 0.0067 | 0.0094 | 5 |
| ResNet34 CIFAR-100 HP-9 | baseline (all Mixup) | – | 79.02 | – | 0.0679 | – | 12 |
| | Ours (a) | 0–6 | 79.28 | +0.26 | 0.0834 | 0.0248 | 8 |
| | | 0–10 | 79.17 | +0.15 | 0.0712 | 0.0971 | 12 |

**Table 7** (continued): Mixup Pause Strategy cross-Model & dataset performance

| Dataset/Model | Method | ENP Duration | Top-1 (%) | Δ | Variance | t-test | Runs |
|---|---|---|---|---|---|---|---|
| ResNet34 CIFAR-100 HP-10 | baseline (all CutMix) | – | 77.76 | – | 0.1088 | – | 16 |
| | Ours (a) | 0–5 | 78.29 | +0.53 | 0.1027 | 0.0052 | 4 |
| | | 0–10 | 78.55 | +0.79 | 0.0640 | 0.0000 | 7 |
| | | 0–15 | 78.45 | +0.69 | 0.0077 | 0.0004 | 4 |
| | | 0–20 | 78.28 | +0.51 | 0.0557 | 0.0047 | 4 |
| ResNet34 CIFAR-100 HP-11 | baseline (all CutMix) | – | 77.72 | – | 0.1591 | – | 16 |
| | Ours (a) | 0–5 | 78.36 | +0.63 | 0.0448 | 0.0036 | 4 |
| | | 0–10 | 78.39 | +0.67 | 0.0626 | 0.0003 | 7 |
| | | 0–15 | 78.35 | +0.63 | 0.0209 | 0.0035 | 4 |
| | | 0–20 | 78.28 | +0.56 | 0.0703 | 0.0087 | 4 |
| Vit-small Tiny-Imagenet HP-12 | vanilla (no Mixup) | – | 41.01 | – | 0.0426 | – | 3 |
| | baseline (all Mixup) | – | 47.09 | – | 0.1118 | – | 4 |
| | Ours (a) | 0–10 | 48.37 | +1.28 | 0.044 | 0.0003 | 3 |
| | | 0–15 | 48.66 | +1.57 | 0.019 | 0.0003 | 3 |
| | Ablation | 0–25 | 46.40 | -0.69 | 0.785 | – | 3 |
| | | 0–35 | 46.79 | -0.30 | 0.400 | – | 3 |
| | | 0–45 | 46.57 | -0.52 | 0.006 | – | 3 |
| | | 30–40 | 46.38 | -0.71 | 0.074 | – | 3 |
| | | 250–270 | 46.60 | -0.49 | 0.018 | – | 3 |
| | | 400–420 | 46.85 | -0.24 | 0.078 | – | 3 |
| Vit-small Tiny-Imagenet HP-13 | vanilla (no Mixup) | – | 41.01 | – | 0.0426 | – | 3 |
| | baseline (all Mixup) | – | 46.04 | – | 0.2457 | – | 7 |
| | Ours (a) | 0–10 | 46.85 | +0.81 | 0.1614 | 0.0128 | 4 |
| | | 0–15 | 46.80 | +0.76 | 0.2165 | 0.0045 | 8 |
| Resnet18 Cifar10 fine-tuning HP-14 | vanilla (no Mixup) | – | 94.80 | – | 0.168 | – | 3 |
| | baseline (all Mixup) | – | 95.51 | – | 0.040 | – | 25 |
| | Ours (a) | 0–6 | 95.61 | +0.10 | 0.069 | 0.2104 | 3 |
| | | 0–10 | 95.61 | +0.10 | 0.031 | 0.0318 | 25 |
| | | 0–15 | 95.48 | -0.03 | 0.012 | 0.4139 | 3 |
| | | 0–20 | 95.52 | +0.02 | 0.044 | 0.4108 | 9 |
| | | 0–25 | 95.52 | +0.02 | 0.032 | 0.4129 | 9 |
| ResNet34 Cifar100 fine-tuning HP-15 | vanilla (no Mixup) | – | 79.47 | – | 1.7491 | – | 3 |
| | baseline (all Mixup) | – | 81.31 | – | 0.0778 | – | 9 |
| | Ours (a) | 0–3 | 81.76 | +0.46 | 0.2760 | 0.0368 | 3 |
| | | 0–6 | 81.32 | +0.01 | 0.1317 | 0.4657 | 9 |
| | | 0–10 | 81.65 | +0.34 | 0.2006 | 0.0339 | 9 |
| | | 0–15 | 81.73 | +0.43 | 0.2565 | 0.0434 | 3 |
| | | 0–20 | 81.90 | +0.60 | 0.2454 | 0.0031 | 9 |
| | Ablation | 0–25 | 81.61 | +0.30 | 0.2266 | – | 3 |
| | | 0–50 | 81.09 | -0.22 | 0.1204 | – | 3 |
| | | 40–60 | 81.18 | -0.12 | 0.1334 | – | 3 |
| | | 100–120 | 81.45 | +0.14 | 0.2071 | – | 3 |
| | | 160–180 | 80.83 | -0.47 | 0.0816 | – | 3 |

*Note:* ENP Duration denotes the length of the Enlightenment Period. HP-n represents different hyperparameter settings, as detailed in Table9.

Table 8: High-Loss Removal Strategy on PreActResNet34 performance

| Dataset/Model | Method | ENP Duration | easy k% | Top-1 (%) | Δ | Variance | t-test | Runs |
|---|---|---|---|---|---|---|---|---|
| PreActResNet34 CIFAR100 | vanilla (no Mixup) | – | – | 75.21 | – | 0.5660 | – | 24 |
| | Ours (a) | 0–1 | 0.75 | 75.37 | +0.16 | 0.1322 | 0.2270 | 14 |
| | | 0–1 | 0.85 | 75.54 | +0.33 | 0.1291 | 0.0153 | 34 |
| | | 0–2 | 0.75 | 75.39 | +0.18 | 0.3850 | 0.3300 | 4 |
| | Ablation | 0–10 | 0.85 | 75.43 | +0.22 | 0.0251 | – | 8 |
| | | 0–20 | 0.85 | 74.69 | −0.85 | 0.7162 | – | 3 |
| | | 0–50 | 0.85 | 74.45 | −1.09 | 1.0267 | – | 3 |
| | | 50–51 | 0.85 | 75.11 | −0.10 | 0.0886 | – | 3 |
| | | 100–101 | 0.85 | 73.77 | −1.64 | 3.1746 | – | 3 |

Table 9: Training hyperparameters in Table7. Ep. = epochs, LR = base learning rate, BS = batch size.

| ID | Model/Dataset | Ep. | $T_{max}$ | LR | Decay | BS | Mixup $\alpha$ |
|---|---|---|---|---|---|---|---|
| HP-1 | PreActResNet34/CIFAR100 | 500 | 160 | 0.002 | 0.1 | 500 | 2 |
| HP-2 | PreActResNet50/CIFAR100 | 500 | 160 | 0.002 | 0.1 | 200 | 2 |
| HP-3 | PreActResNet50/Tiny-Imagenet | 500 | 160 | 0.002 | 0.1 | 220 | 2 |
| HP-4 | PreActResNet34/CIFAR100 | 500 | 160 | 0.002 | 0.1 | 500 | 2 |
| HP-5 | PreActResNet18/FOOD101 | 120 | 100 | 0.005 | 0.004 | 128 | 2 |
| HP-6 | ResNet34/CIFAR100 (v1) | 200 | 160 | 0.002 | 0.004 | 500 | 2 |
| HP-7 | ResNet34/CIFAR100 (v1) | 200 | 160 | 0.002 | 0.004 | 500 | 1 |
| HP-8 | ResNet34/CIFAR100 (v2) | 250 | 70 | 0.005 | 0.009 | 1000 | 2 |
| HP-9 | ResNet34/CIFAR100 (v2) | 250 | 70 | 0.005 | 0.009 | 1000 | 1 |
| HP-10 | ResNet34/CIFAR100 (v2) | 250 | 70 | 0.005 | 0.009 | 1000 | 1 |
| HP-11 | ResNet34/CIFAR100 (v2) | 250 | 70 | 0.005 | 0.009 | 1000 | 2 |
| HP-12 | ViT-small/Tiny-Imagenet | 500 | 160 | 0.0003 | 0.0001 | 200 | 2 |
| HP-13 | ViT-small/Tiny-Imagenet | 500 | 160 | 0.0003 | 0.0001 | 200 | 1 |
| HP-14 | ResNet18-CIFAR10 (fine-tune) | 200 | 60 | 0.001 | 0.1 | 200 | 2 |
| HP-15 | ResNet34-CIFAR100 (fine-tune) | 200 | 60 | 0.001 | 0.1 | 200 | 2 |

Table 10: Our training recipe on ViT/T-ImageNet, adapted from Liu et al. (2023).

| Training Setting | Configuration |
|---|---|
| weight init | trunc. normal (0.2) |
| optimizer | AdamW |
| base learning rate | 4e-3 |
| weight decay | 0.05 |
| optimizer momentum | $\beta_1, \beta_2{=}0.9, 0.999$ |
| batch size | 4096 |
| training epochs | 300 |
| learning rate schedule | cosine decay |
| warmup epochs | 50 |
| warmup schedule | linear |
| stochastic depth rate Huang et al. (2016) | 0.0 |
| dropout rate Hinton et al. (2012) | 0.0 |
| randaugment Cubuk et al. (2020) | (9, 0.5) |
| Mixup Zhang et al. (2018) | 0.8 |
| cutmix Yun et al. (2019) | 1.0 |
| random erasing Zhong et al. (2020) | 0.25 |
| label smoothing Szegedy et al. (2016) | 0.1 |
| layer scale Touvron et al. (2021) | 1e-6 |
| gradient clip | None |
| exp. mov. avg. (EMA) Polyak & Juditsky (1992) | None |

