# OpenReview forum: "Enlightenment Period Improving DNN Performance"
_ICLR.cc/2026/Conference — Submitted to ICLR 2026_

### Official Review · Reviewer_iEY9 · 2025-10-28

**Soundness:** 1
**Presentation:** 2
**Contribution:** 2
**Rating:** 2
**Confidence:** 3

**Summary:**

This paper introduces the "Enlightenment Period" (ENP), defined as the initial training phase of a deep neural network from the start until accuracy reaches approximately 50%. The authors posit that during this period, Mixup data augmentation has a dual effect: a detrimental "Gradient Interference Effect" and a beneficial "Activation Revival Effect". They further theorize that this negative interference diminishes with increased model or data size, which they term the "Gradient Interference Diminishing Effect". Based on this trade-off, the paper proposes three strategies: the Mixup Pause Strategy for small-scale scenarios , the Alpha Boost Strategy for large-scale, underfitting models , and the High-Loss Removal Strategy for tasks where Mixup is inapplicable (e.g., time-series, LLMs). The authors claim these strategies achieve superior performance across diverse models like ViT and ResNet on datasets including CIFAR and ImageNet-1K.

**Strengths:**

1. The paper attempts to analyze the very early, chaotic phase of training and connect it to practical, data-driven strategies.
2. The investigation of these strategies (or heuristics) is applied across multiple domains, including CV, time-series, and LLMs, which shows breadth.

**Weaknesses:**

1. **Oversimplified Theoretical Model:** The core theoretical analyses of "Gradient Interference" (Section 2.2) and the "Diminishing Effect" (Section 2.3) are based on a simple linear model with a sigmoid activation ($\hat{y}=\sigma(\theta^{\top}x)$). This model does not capture the non-linear dynamics, layer interactions, or high-dimensional parameter spaces of modern ResNets and Vision Transformers. Conclusions drawn from this toy model (e.g., about gradient orthogonality) cannot be reasonably extrapolated to these deep architectures.
2. **Marginal and Unconvincing Empirical Gains:** The reported performance improvements are modest (e.g., +0.10% on CIFAR-10, +0.4% on ImageNet-1K). Such minimal gains do not provide a compelling reason to adopt these strategies, especially given their added complexity.
3. **Introduction of a Confounding Hyperparameter:** The proposed strategies introduce a new, highly sensitive hyperparameter: the "ENP Duration"15. Ablation studies in Table 3 and Table 7 show that an incorrect choice for this duration can degrade performance, sometimes significantly (e.g., ViT-S/Tiny-ImageNet performance drops from +1.57% to -0.69% when changing duration from 10 to 25 epochs). This makes the method impractical, as it merely replaces the tuning of one hyperparameter (Mixup $\alpha$) with another, less-understood one (ENP duration).
4. **Weak Evidence for the Activation Revival Effect:** The "Activation Revival Effect" (Section 2.4) is supported only by correlational evidence. Figure 4 shows that Mixup training results in fewer zero-value activations. This is an expected property of Mixup (which interpolates samples, pulling them away from saturated decision boundaries) and does not prove a causal "revival" mechanism that improves performance.
5. **Writing and Presentation:** The overall writing of this paper is not good. Some figures are difficult to interpret. The 2D embeddings in Figure 1 and Figure 7 are low-resolution and do not provide clear, quantitative evidence for the claims about "disorganized clusters" or "boundary-ambiguous characteristics".

**Questions:**

1. **Order parameter:** In the introduction, the authors frame the "Enlightenment Period" (ENP) using phase transition theory and suggest that accuracy acts as a "quasi-order parameter." However, this concept is not explained in detail or utilized in the subsequent theoretical analysis (Section 2). Can the authors clarify the role of the "order parameter" concept in your work? How does this analogy concretely inform the development of your theoretical effects (e.g., Gradient Interference) or practical strategies?

2. **ENP Duration:** A critical, practical aspect of this work is determining the duration of the ENP. The authors suggest it ends "around the point when the model accuracy reaches 50%", but the experiments (e.g., Table 3, Table 7) appear to be a grid search over a number of epochs (e.g., 10, 15, 20).
- How is the optimal ENP duration determined in practice? Is it a hyperparameter that must be tuned via a full grid search for every new model/dataset pair?
- If the authors do use the 50% accuracy criterion, what is the performance? For instance, in the experiments, what epoch corresponds to ~50% accuracy, and how does applying the "Mixup Pause" strategy for that specific duration compare to the optimal duration grid search?

3. **Assumption:** In Section 2.2 (L146), the derivation of the Mixup gradient relies on the assumption that $|f_i| > |f_j|$ (where $x_i$ is a positive sample and $x_j$ is a negative sample). Why should this be the case during early training, when the model's classifications are essentially random and scores are likely to be small and noisy for both classes?

4. **Expectation of the gradient perturbation:** In Section 2.3.1 (L205), the authors assume that the expectation of the gradient perturbation, $\mathbb{E}[\delta^{(k)}]$, is approximately zero. Why is it the case?

5. **Experimental validation:** In the experimental validation (Section 2.3.3), the authors define the gradient interference ratio $r$ using only the component of the Mixup gradient that is perpendicular to the vanilla gradient. This seems problematic. First, the mixup gradient is not entirely orthogonal as shown in Figure 2. Second, the norm of perpendicular component is usually smaller than that of the entire gradient, leading to a small value of $r$.

6. **Small/large scale:** The authors propose a "boundary" between small-scale and large-scale scenarios (L336), which you define as "ViT-T when trained on the 100% ImageNet-1K dataset." This seems arbitrary.
- Can you provide a more principled justification for this boundary?
- Out of your 16 experimental settings, it appears only *one* (ViT-T/ImageNet-1K) is considered "large-scale," and all others are "small-scale." Can you confirm this?
- If so, how can you be confident in the "Alpha Boost" strategy when it has only been validated in a single experimental setting? Would it not be necessary to test this on other large-scale setups (e.g., larger models than ViT-T or larger datasets than ImagetNet-1K) to validate the claim?

7. **Missing related works:** The High-Loss Removal strategy shares conceptual similarities with recent work on simplicity bias [1], which also identifies an early training phase to find hard samples and modify the data distribution. Can the authors please discuss the relationship between "High-Loss Removal" strategy and their approach? How does your ENP definition and method differ from their use of early-phase checkpoints to mitigate simplicity bias?

8. **Theoretical proofs:** 2. The derivations for BENR (Appendix C.1) and ATD (Appendix C.2) appear to be based on non-rigorous examples.
- The BENR derivation (Eq. 18-26) is based on a single concrete example of two positive samples and one negative sample. A proof by example is not rigorous. Can the authors provide a more general derivation for $BENR_{vanilla} > BENR_{mix}$?
- In Appendix C.2, how to transition from Equation (28) (which compares the L2 norm of the *parameter updates*) to Equation (29) (which compares the *variance* of the *activation trajectory distance*)? Please provide the missing steps.

9. **Alternative choice of distance:** The authors use Activation Trajectory Distance (ATD), defined as the L2 norm of the difference in activation vectors, to quantify representation change. Given that these are high-dimensional representations, L2 distance can be a noisy and potentially misleading metric. Have you considered using more robust, alignment-based similarity metrics like Centered Kernel Alignment (CKA) [2] to verify your findings about representation dynamics?

[1] Nguyen, Tuan H., et al. "Changing the training data distribution to reduce simplicity bias improves in-distribution generalization." Advances in Neural Information Processing Systems 37 (2024): 68854-68896
[2] Kornblith, Simon, et al. "Similarity of neural network representations revisited." International conference on machine learning. PMlR, 2019.

---

> ### Author Response · Authors · 2025-11-19
> **1. Paper Summary**
>
> We sincerely appreciate the reviewers for their comments on our manuscript. We have carefully read and thoroughly addressed each comment. Due to word count constraints, our responses are divided into several comments for each reviewer:
>
> **1. Paper Summary**
>
> **2. Supplementary Results: Alpha Boost on ImageNet-1K**
>
> **3. Theoretical Reasoning for the ~50% Accuracy Endpoint of the Enlightenment Period**
>
> **4. Addressing Weaknesses and Questions**
>
>
> ### Paper Summary
> Given the density of the theoretical derivations and the number of novel components introduced in our work, we provide a concise summary of the paper’s core contributions as follows:
>
> This paper identifies a critical "Enlightenment Period (ENP)" in the early stage of deep neural network training. As can be observed from Figure 1, the Enlightenment Period undergoes a process from disorder to order. Drawing on phase transition theory in physics, we use classification accuracy as a "quasi-order parameter".
>
> ***Based on a mathematical model, we derive and prove the following conclusions:***
>
> 1. In the early stage of training, when the "quasi-order parameter" (ACC) is very low, the Mixup data augmentation technique induces the Gradient Interference Effect during this period (Equations 1–3).As training progresses to the middle and late stages, with the increase of ACC, the model gradually transitions to an ordered state, and Mixup then plays a role in optimizing the decision boundary (Lines 178–183).
>
> 2. Meanwhile, we further prove that this Gradient Interference Effect diminishes with the increase in sample size ( Oₚ(1/√N)) and the expansion of parameter size (Oₚ(1/√D)) (Section 2.3).
>
> 3. In Section 2.4, we innovatively propose an optimization effect of Mixup: the Activation Revival Effect. Mixup can pull some activation values in the negative interval back to the differentiable interval, thereby restoring gradient propagation. Therefore, during the ENP, Mixup exhibits two opposing effects:"interference" and "revival". Their dominance depends on the scale of the model and data size.
>
> **All the results above have been also verified by experiments (Figures 2, 3, and 4). The validation of the modeling rationality is provided in Appendix C.**
>
> Furthermore, we find that high-loss samples share similar characteristics with Mixup samples, which has been verified by the experiments in Figure 7 and the proofs in Appendix B.
>
> Based on the above theories, we propose three scenario-specific optimized training strategies:
>
> - Mixup Pause Strategy (suitable for small-scale scenarios to avoid gradient interference);
> - Alpha Boost Strategy (suitable for large-scale underfitting models to enhance the Activation Revival Effect);
> - High-Loss Sample Removal Strategy (suitable for domains where Mixup is inapplicable, such as LLMs and time series tasks).
>
> Tables 3, 4, and 5 in the main text of this paper, as well as Table 7 in the appendix, present large-scale verification **results covering 180+ parameter combinations, involving various models (ResNet, ViT, LLMs) and datasets (CIFAR, ImageNet, etc.). All core performance improvements are verified by one-tailed t-tests (p<0.05)**, and ablation experiments confirm that these strategies are only effective during the ENP (Table 4,7), fully verifying the effectiveness and generalizability of the proposed methods.

---

> ### Author Response · Authors · 2025-11-19
> **2. Supplementary Results: Alpha Boost on ImageNet-1K**
>
> After the submission of this paper, we have continued our experiments and supplemented the following results to demonstrate the broad effectiveness of alpha-boost in large-scale scenarios. Each result represents the average value obtained from at least 3 random seeds. The parameters are consistent with those in Table 10 of the Appendix. All experiments were conducted on an 8-GPU H100, with a single run time ranging from 9 to 15 hours.
> | Strategy | Model/Dataset |  ENP $\alpha$ | ENP Dur | Top-1 Acc. (%)  ours | Baseline | Δ |
> ------------------|--------|--------|--------|----------------|-------|-------|
> | Alpha Boost | ViT-T/ImageNet-1K |  2 | 20 E | 74.3±0.16 | 73.9±0.20 | +0.4 |
> | Alpha Boost  | ViT-S/ImageNet-1K |  1.5 | 20 E | 80.7±0.23 | 80.3±0.19 | +0.4 |
> | Alpha Boost  | ResNet50/ImageNet-1K |  1.5 | 20 E | 79.2±0.29 | 78.7±0.25 | +0.5|
> | Alpha Boost  | Convnext-F/ImageNet-1K |  2 | 10 E | 76.3±0.27 | 76.1±0.22 | +0.2|
>
> The following tables are the detailed data and ablation results of the three newly supplemented models：
>
> #### Vit_Small
> | ENP α \ ENP Dur | 15     | 20     | 25     | 50 (ablation) |
> |------------------|--------|--------|--------|----------------|
> | 0.8 (baseline)   | 80.34  | 80.34  | 80.34  | 80.34          |
> | 1                | 80.42  | 80.52  | 80.50  | —              |
> | 1.5              | 80.35  | **80.70**  | 80.58  | 80.02          |
> | 2                | 80.49  | 80.63  | 80.46  | —              |
> #### ResNet50
> |  ENP α\ ENP Dur    | 15      | 20   | 25   | 50 (ablation) |
> |---------------|------------------------|-----------|-------------|-----------|
> | 0.8 (baseline) | 78.70  | 78.70 | 78.70 | 78.70 |
> | 1.2   | 78.84   | 79.09 | 78.80 | — |
> | 1.5   | 78.92   | **79.22** | 78.93 | 78.31 |
> | 2     | 78.86   | 79.08 | 78.64 | — |
>
> #### Convnext_femto
> | ENP α \ ENP Dur | 6      | 10     | 15     | 50 (ablation) |
> |------------------|--------|--------|--------|----------------|
> | 0.8 (baseline)   | 76.11   | 76.11  | 76.11  | 76.11          |
> | 1.5              | 75.95  | 76.16  | 75.97  | —              |
> | 2                | 76.10  | **76.32**  | 75.92  | 75.22          |
> | 2.5              | 75.97  | 75.83  | 75.88  | —              |

---

> ### Author Response · Authors · 2025-11-19
> **3. Theoretical Reasoning for the ~50% Accuracy Endpoint of the Enlightenment Period**
>
> ### Theoretical Reasoning for the ~50% Accuracy Endpoint of the Enlightenment Period
> To address the reviewers’ concerns, we provide a theoretical derivation indicating that the Enlightenment Period ends when the accuracy reaches about 50%.
>
> We make the following assumptions:
> 1. For a multi-class classification task with $k \geq 2$ classes ($k \geq 2$), the class distribution is balanced (prior probability of $1/k$ for each class) and samples are independent and identically distributed (i.i.d.).
> 2. The model’s discriminative ability is characterized by a scalar $t$: for any given sample, the model classifies it correctly with probability $t$ and incorrectly with probability $1-t$.
> 3. When making incorrect predictions, the model’s guessing behavior is approximately unbiased across all classes; under the same network architecture and training configuration, the critical discriminative power $t$ corresponding to the "end of the Enlightenment Period" remains consistent between binary and multi-class tasks.   Under these assumptions, consider a binary classification task ($k=2$): the model’s predictions fall into two scenarios:
> - Correct prediction (probability $t$) contributes a correct probability of $t \cdot 1$;
> - Incorrect prediction (probability $1-t$): random guessing between the two classes with an accuracy of $1/2$, contributing a correct probability of $(1-t)\cdot \tfrac{1}{2}$.
> By the law of total probability, the overall accuracy $X$ of the binary classification task is:   $$X = t \cdot 1 + (1-t)\cdot \frac{1}{2}$$   Rearranging gives the relationship between discriminative power and binary classification accuracy:   $$t = 2X - 1. \tag{1}$$   For a general $k$-class classification task, the prediction logic is consistent with the binary case:
> - Correct prediction (probability $t$) contributes a correct probability of $t \cdot 1$;
> - Incorrect prediction (probability $1-t$): unbiased random guessing across $k$ classes, contributing a correct probability of $(1-t)\cdot \tfrac{1}{k}$.
> Thus, the overall accuracy $P(k)$ of the $k$-class task at discriminative power $t$ is:   $$P(k) = t \cdot 1 + (1-t)\cdot \frac{1}{k}. \tag{2}$$   Substituting $t = 2X - 1$ (derived from binary classification accuracy in Eq. (1)) into Eq. (2):   $$P(k) = 2X - 1 + \frac{2(1-X)}{k}$$   Taking the binary classification experiment in Fig. 2 of the original paper as an example: the accuracy of the binary task at the end of the Enlightenment Period is approximately $X \approx 0.75$. From Eq. (1), the critical discriminative power at this point is:   $$t \approx 2\times 0.75 - 1 = 0.5$$   Under the same network architecture and optimization configuration, the critical discriminative power $t$ corresponding to the "end of the Enlightenment Period" can be approximated as invariant across different $k$. Substituting $X \approx 0.75$ into the above expression:   $$P(k) \approx 2\times 0.75 - 1 + \frac{2(1-0.75)}{k} = 0.5 + \frac{0.5}{k}$$   This shows that under the two-state competence model and constant discriminative power assumption, when the binary classification accuracy at the end of the Enlightenment Period is approximately $X \approx 0.75$, the corresponding accuracy of the $k$-class task at the end of the Enlightenment Period naturally falls around $0.5 + 0.5/k$. For typical multi-class datasets with $k \geq 10$ in this paper, this value converges to around 50%, which is quantitatively consistent with the empirical phenomenon observed in multiple experiments (Table 2 in our paper) that "the Enlightenment Period ends around 50% accuracy.

---

> > ### Author Response · Authors · 2025-11-19
> > **4. Addressing Weaknesses and Questions_1**
> >
> > ## Response to Weakness
> > 1.**Oversimplified Theoretical Model:The core theoretical analyses of "Gradient Interference" (Section 2.2) and the "Diminishing Effect" (Section 2.3) are based on a simple linear model with a sigmoid activation (). This model does not capture the non-linear dynamics, layer interactions, or high-dimensional parameter spaces of modern ResNets and Vision Transformers. Conclusions drawn from this toy model (e.g., about gradient orthogonality) cannot be reasonably extrapolated to these deep architectures.**
> >
> > Response:In the field of DNN training dynamics, numerous prior studies (see Appendix A.1) ground their theoretical derivations and metrics in **simplified analytical models**. This is because mathematically modeling the dynamics of intricate nonlinear functions remains extremely challenging given the current state of human mathematical capabilities. Although we cannot accurately characterize complex models in practical scenarios that are defined by large parameter sizes and intricate architectures, there is a prevalent paradigm in existing research: deriving conclusions using linear or simplified models, then extrapolating these properties to complex models, which consistently delivers performance improvements. Notable examples include Mixup and Manifold Mixup. As explicitly stated in the original Mixup paper, “linearity works effectively” for such extrapolation.
> >
> > More importantly, we provide three independent lines of evidence demonstrating that the Enlightenment Period model introduced in Sections 2.1–2.2 is well-founded and broadly applicable to general DNN training.
> >
> > First, the conclusions in Sections 2.2–2.4 have been validated through multi-perspective experimental results (see Figures 2, 3, and 4).
> >
> > Second, based on the modeling assumptions and key conclusions established in Sections 2.1–2.2 (e.g. high-dimensional normalization, sample orthogonality, and the gradient deviation formulation), we derived the early-stage trends of BENR and ATD, which match the empirical observations precisely. BENR and ATD therefore serve not only as indicators of Enlightenment Period behavior, but also as quantitative validators of the correctness of our theoretical model (see Appendix C).
> >
> > Third, strategies (in Section 2.5)derived directly from the theoretical results yield statistically significant performance improvements (p < 0.05) across diverse datasets and model architectures.
> >
> > 2.**Marginal and Unconvincing Empirical Gains: The reported performance improvements are modest (e.g., +0.10% on CIFAR-10, +0.4% on ImageNet-1K). Such minimal gains do not provide a compelling reason to adopt these strategies, especially given their added complexity.**
> >
> > Response:
> > - **In the field of optimization, performance gains from model-training parameters are inherently limited today, with most improvements hovering around 0.4% [1]**. Notably, our baseline for ImageNet-1K integrates optimal parameter combinations derived from various SOTA training strategies available today.
> > - **Our code modifications are minimal** and do not involve any changes to model architectures.
> > - The 0.1% improvement on CIFAR-10 (fine-tuning) is the smallest gain reported in our entire paper. We have faithfully documented all experimental results across CIFAR-10, CIFAR-100, Tiny-ImageNet, ImageNet-1K, large language models (LLMs), and time-series tasks, including performance improvements, variances, and repeated runs. All reported improvements have passed the p-test, confirming their statistical significance. While experimental results inevitably exhibit fluctuations, the 180+ experimental setups in our paper sufficiently validate the widespread existence of the Enlightenment Period phenomenon.
> > - Furthermore, the primary goal of our research is not to pursue state-of-the-art (SOTA) performance on each datasets. Instead, **its greater significance lies in the discovery of the unique Enlightenment Period phenomenon, which offers a novel perspective for investigating the optimization process of DNN training**.
> > [1]Salehin, I. and Kang, D.K., 2023. A review on dropout regularization approaches for deep neural networks within the scholarly domain. Electronics, 12(14), p.3106.

---

> > > ### Author Response · Authors · 2025-11-19
> > > **4. Addressing Weaknesses and Questions_2**
> > >
> > > 3.**Introduction of a Confounding Hyperparameter:The proposed strategies introduce a new, highly sensitive hyperparameter: the "ENP Duration"15. Ablation studies in Table 3 and Table 7 show that an incorrect choice for this duration can degrade performance, sometimes significantly (e.g., ViT-S/Tiny-ImageNet performance drops from +1.57% to -0.69% when changing duration from 10 to 25 epochs). This makes the method impractical, as it merely replaces the tuning of one hyperparameter (Mixup ) with another, less-understood one (ENP duration).**
> > >
> > > The "ENP Duration" hyperparameter you mentioned is the core focus of this study. We have supplemented the theoretical proof on how to determine the ENP Duration. Combined with the empirical evidence from Table 4 in the original paper and the explanation of "quasi-order parameter" in the introduction, these contents will facilitate your better understanding of this core concept.
> > > Additionally, commonly used hyperparameters in model training, such as learning rate, dropout rate, and warm-up duration, are equally sensitive; even minor adjustments can significantly impact training outcomes. The new training strategy proposed in this paper is no more "sensitive" than these well-established hyperparameters.
> > > The core focus of this paper lies in identifying a unique pattern in the training process of neural networks, and the explanations of phase transition theory and "quasi-order parameter" in the introduction facilitate a better understanding of this brief yet distinctive Enlightenment Period.
> > > Furthermore, we aim to clarify a misunderstanding: the Enlightenment Period Duration and Mixup-α are entirely distinct hyperparameters, and we do not replace Mixup-α with the ENP Duration. As detailed in the strategies presented in Sections 2.2-2.5 of the paper, performance improvements are achieved by adjusting Mixup-α specifically during the Enlightenment Period, while the value of Mixup-α after the Enlightenment Period remains consistent with that in existing studies.
> > >
> > > 4.**Weak Evidence for the Activation Revival Effect: The "Activation Revival Effect" (Section 2.4) is supported only by correlational evidence. Figure 4 shows that Mixup training results in fewer zero-value activations. This is an expected property of Mixup (which interpolates samples, pulling them away from saturated decision boundaries) and does not prove a causal "revival" mechanism that improves performance.**
> > >
> > > Response:
> > > Our "Activation Revival Effect" is not solely based on correlational evidence but supported by explicit analysis and experiments. Figure 4 not only shows a significant reduction in zero-value activations under Mixup training but also forms a causal loop through the performance improvement of the Alpha Boost strategy: increasing the alpha value of Mixup enhances this effect, and this enhancement is strictly positively correlated with the model's performance improvement. As evidenced by results in scenarios such as ViT-T on ImageNet-1K (Table 3), this directly proves its optimization effect on performance. This mechanism and the gradient interference and decision boundary optimization derived in Section 2.2 **are two completely independent pathways**. The interpolation pulling samples away from saturated decision boundaries" you mentioned corresponds to Mixup's role in boundary refinement, which has been elaborated in Section 2.2. In contrast, the Activation Revival Effect focuses on the gradient vanishing problem caused by neuron saturation, restoring the gradient update of saturated neurons through sample mixing. To the best of our knowledge, this optimization mechanism has not been proposed before and is distinct from the boundary changes brought by interpolation.
> > >
> > > 5.**Writing and Presentation:The overall writing of this paper is not good. Some figures are difficult to interpret. The 2D embeddings in Figure 1 and Figure 7 are low-resolution and do not provide clear, quantitative evidence for the claims about "disorganized clusters" or "boundary-ambiguous characteristics".**
> > >
> > > Response:
> > > All figures in our paper are high-resolution vector graphics (which may take a short time to load). If you could wait patiently for 5-6 seconds, you will see clearer and more interpretable visualizations.

---

> > > > ### Author Response · Authors · 2025-11-19
> > > > **4. Addressing Weaknesses and Questions_3**
> > > >
> > > > ## Response to Questions
> > > >
> > > > 1.**Order parameter: In the introduction, the authors frame the "Enlightenment Period" (ENP) using phase transition theory and suggest that accuracy acts as a "quasi-order parameter." However, this concept is not explained in detail or utilized in the subsequent theoretical analysis (Section 2). Can the authors clarify the role of the "order parameter" concept in your work? How does this analogy concretely inform the development of your theoretical effects (e.g., Gradient Interference) or practical strategies?**
> > > >
> > > > Response:
> > > > Treating accuracy as a "quasi-order parameter" is a foundational assumption of our paper’s mathematical model, as detailed in Section 2.1 (Modeling Setup). Specifically, when accuracy is low (early training), Mixup interferes with gradient descent; as accuracy rises, it instead enhances model generalization. As noted in the introduction, we analogize the complex training dynamics of DNNs in early stages, and we particularly focus on the shift from disorganized to organized representations, to phase transitions in physics, and elaborate in the methodology that this DNN "phase transition" is driven by increasing accuracy and decreasing misclassification rates.
> > > >
> > > >
> > > > 2.**ENP Duration: A critical, practical aspect of this work is determining the duration of the ENP. The authors suggest it ends "around the point when the model accuracy reaches 50%", but the experiments (e.g., Table 3, Table 7) appear to be a grid search over a number of epochs (e.g., 10, 15, 20).
> > > >
> > > > Response:
> > > > Details regarding the Enlightenment Period  Duration can be found in the theoretical deductions above and the experiments presented in our paper. As we analyzed earlier, the ENP concludes around the 50% accuracy mark, so it is not confined to a fixed number of epochs. For practical guidance on determining the ENP Duration in real-world training, please refer to Lines 355-359 of the original manuscript, where specific operational instructions are provided.
> > > >
> > > > 3.**Assumption: In Section 2.2 (L146), the derivation of the Mixup gradient relies on the assumption that  (where  is a positive sample and  is a negative sample). Why should this be the case during early training, when the model's classifications are essentially random and scores are likely to be small and noisy for both classes?**
> > > >
> > > > The assumption that "for positive samples, $f_i = \theta^\top x_i < 0$; for negative samples, $f_j = \theta^\top x_j > 0$; and $|f_i| > |f_j|$" in the Mixup gradient derivation (Section 2.2, L146-148) is justified based on the core characteristics of the Enlightenment Period(acc is very low). The Enlightenment Period is defined as the early training stage where the model’s accuracy remains<50%, and its classification capability is inherently weak. Randomly initialized parameters are far from the optimal solution, leading to "high-confidence misclassifications" rather than random, low-magnitude scores. For positive samples, the parameter vector $\theta$ aligns oppositely with the sample features $x_i$, resulting in $f_i = \theta^\top x_i < 0$; for negative samples, $\theta$ aligns with $x_j$, leading to $f_j = \theta^\top x_j > 0$. As for $|f_i| > |f_j|$ , it stems from the disordered parameter state in early training. Without effective feature learning, the model’s linear score function produces extreme values for misclassified samples which can be far from the decision boundary (where $f = 0$). This is not a "small and noisy" scenario but a direct consequence of random initialization, which is widely observed in DNN training [1,2].
> > > > Besides, as verified in Figure 2 of the paper, early training shows significant misalignment between Mixup and vanilla gradients, which relies on the model’s extreme misclassification scores. This experimental phenomenon corroborates the rationality of the assumption.
> > > >
> > > > [1] Guo, C., Pleiss, G., Sun, Y., & Weinberger, K. Q. (2017). On Calibration of Modern Neural Networks. *ICML*.
> > > > [2] Zhu, L., et al. (2022). Rethinking Confidence Calibration for Failure Prediction. *ECCV*.

---

> > > > > ### Author Response · Authors · 2025-11-19
> > > > > **4. Addressing Weaknesses and Questions_4**
> > > > >
> > > > > 4.**Expectation of the gradient perturbation:In Section 2.3.1 (L205), the authors assume that the expectation of the gradient perturbation $\mathbb{E}[\boldsymbol\delta^{(k)}]$ is approximately zero. Why is it the case?**
> > > > >
> > > > >   For a fixed sample pair $(x_i^+, x_j^-)$, the Mixup gradient deterministically deviates from the vanilla direction by $\boldsymbol\delta = -1/4(x_i + x_j)$, and this single perturbation is indeed non-zero-mean. However, our assumption $\mathbb{E}[\boldsymbol\delta^{(k)}] \approx 0$ is grounded in **random sampling of Mixup pairs across training steps**, not fixed pairs. At each training step $k$, the sample pair $(x_i^{(k)}, x_j^{(k)})$ is randomly and independently drawn from the training set.   Mixup randomly pairs positive and negative samples, leading to random variation of the $\pm$ sign in Eq. (4).  Thus, the zero expectation holds when averaging over all possible random samples, derived as follows:   $$ \begin{align*} \mathbb{E}_k[\boldsymbol\delta^{(k)}] &= \mathbb{E}_k\left[\pm \frac{1}{4}(x_i^{(k)} + x_j^{(k)})\right] \\ &= \frac{1}{4} \cdot \mathbb{E}_k\left[\pm (x_i^{(k)} + x_j^{(k)})\right] \end{align*} $$   Under balanced training data and uniform random sampling:   Positive sample $x_i^+$ pairs with negative sample $x_j^-$, giving $\boldsymbol\delta \propto +(x_i^+ + x_j^-)$;  Negative sample $x_i^-$ pairs with positive sample $x_j^+$, giving $\boldsymbol\delta \propto -(x_i^- + x_j^+)$.   Let $\mu_+$ and $\mu_-$ denote the class means of positive and negative samples, respectively(Lines188 in our paper). Substituting into the expectation:   $$ \mathbb{E}_k[\boldsymbol\delta^{(k)}] \approx \frac{1}{2} \cdot \left(+\frac{1}{4}(\mu_+ + \mu_-)\right) + \frac{1}{2} \cdot \left(-\frac{1}{4}(\mu_- + \mu_+)\right) = 0 $$ This zero expectation is valid **across random batch sampling**, not for any individual sample pair. Each single Mixup sample introduces non-zero interference (as analyzed in Section 2.2), but these perturbations behave as **unbiased noise** when averaged over the entire training set.
> > > > >
> > > > > 5.**Experimental validation:In the experimental validation (Section 2.3.3), the authors define the gradient interference ratio  using only the component of the Mixup gradient that is perpendicular to the vanilla gradient. This seems problematic. First, the mixup gradient is not entirely orthogonal as shown in Figure 2. Second, the norm of perpendicular component is usually smaller than that of the entire gradient, leading to a small value of r.**
> > > > >
> > > > >
> > > > > Response:
> > > > > The claim that "Mixup gradients are not perfectly orthogonal" does not contradict our setup. Figure 2 merely demonstrates that there are a large number of sample pairs with large gradient angles during the Enlightenment Period, rather than strictly 90-degree angles. Therefore, we explicitly isolate the perpendicular component through projection operations, which is precisely to avoid assuming "perfect orthogonality". On the other hand, the fact that "the norm of the perpendicular component is usually smaller than that of the entire gradient" is not a flaw: the purpose of the gradient interference ratio (r) is not to prove that the absolute value of interference is large, but to capture the "interference-to-useful-signal ratio". As verified in Figure 3, this ratio decreases significantly with the increase in sample size or model parameter scale, which perfectly aligns with the theoretical expectation of gradient interference attenuation. This confirms that our definition of r can effectively characterize the core law, and there is no logical inconsistency.
> > > > >
> > > > > 6. **Small/large scale:The authors propose a "boundary" between small-scale and large-scale scenarios (L336), which you define as "ViT-T when trained on the 100% ImageNet-1K dataset." This seems arbitrary.- Can you provide a more principled justification for this boundary?Out of your 16 experimental settings, it appears only one (ViT-T/ImageNet-1K) is considered "large-scale," and all others are "small-scale." Can you confirm this?If so, how can you be confident in the "Alpha Boost" strategy when it has only been validated in a single experimental setting? Would it not be necessary to test this on other large-scale setups (e.g., larger models than ViT-T or larger datasets than ImagetNet-1K) to validate the claim?**
> > > > >
> > > > >
> > > > > Response:
> > > > > Supplementary large-scale experiments are provided above. Detailed information referenced to the performance data across different ImageNet data scales (i.e., ViT-T trained on ImageNet1K) as presented in Table 1.

---

> > > > > > ### Author Response · Authors · 2025-11-19
> > > > > > **4. Addressing Weaknesses and Questions_5**
> > > > > >
> > > > > > 7.**Missing related works: The High-Loss Removal strategy shares conceptual similarities with recent work on simplicity bias [1], which also identifies an early training phase to find hard samples and modify the data distribution. Can the authors please discuss the relationship between "High-Loss Removal" strategy and their approach? How does your ENP definition and method differ from their use of early-phase checkpoints to mitigate simplicity bias?**
> > > > > >
> > > > > > Response:
> > > > > >
> > > > > > our High-Loss Removal strategy is conceptually different in several respects. First, our method is restricted to the Enlightenment Period, a very short stage from the start of training until accuracy reaches about 50, whereas simplicity bias methods typically use early checkpoints only to design sampling or weighting schemes that persist throughout the entire training trajectory. Second, our objective is not to correct a simplicity bias in the final solution but to facilitate the disordered to ordered phase transition by temporarily reducing gradient interference during this critical period, after which training proceeds in a standard way. Third, the role of samples is inverted: in our view, high loss samples are boundary ambiguous but still valuable in later epochs, so we remove them only during the Enlightenment Period and then fully restore them, while simplicity bias approaches treat easy samples as potentially harmful or redundant and keep suppressing them over the whole course of training. Fourth, the research method proposed in Paper [1] is exclusively applicable to CNNs and requires the separation and clustering of early-stage training samples. It differs fundamentally from our work in terms of research objectives and methodologies, and the mathematical models of the two papers are also completely distinct.
> > > > > >
> > > > > >
> > > > > > 8 **Theoretical proofs:The derivations for BENR (Appendix C.1) and ATD (Appendix C.2) appear to be based on non-rigorous examples.
> > > > > > The BENR derivation (Eq. 18-26) is based on a single concrete example of two positive samples and one negative sample. A proof by example is not rigorous. Can the authors provide a more general derivation for ?**
> > > > > >
> > > > > > Response:
> > > > > > One positive sample and one negative sample are mixed into a 50%-50% Mixup sample, which is combined with another positive sample. This setup is used to estimate scenarios where the mixing ratio ranges between 0 and 50%.
> > > > > >
> > > > > > **In Appendix C.2, how to transition from Equation (28) (which compares the L2 norm of the parameter updates) to Equation (29) (which compares the variance of the activation trajectory distance)? Please provide the missing steps.**
> > > > > >
> > > > > > Response:
> > > > > > Because in vanilla training, the magnitude of each parameter update (quantified by the L2 norm of Δθ_vanilla) is greater than that in Mixup training (quantified by the L2 norm of Δθ_mix), this greater parameter update magnitude in vanilla training in turn leads to more instability in its activation variations. This causal chain ultimately results in Equation (29).

---

> > > > > > > ### Author Response · Authors · 2025-11-19
> > > > > > > **4. Addressing Weaknesses and Questions**
> > > > > > >
> > > > > > > 9. **Alternative choice of distance:** The authors use Activation Trajectory Distance (ATD), defined as the L2 norm of the difference in activation vectors, to quantify representation change. Given that these are high-dimensional representations, L2 distance can be a noisy and potentially misleading metric. Have you considered using more robust, alignment-based similarity metrics like Centered Kernel Alignment (CKA) [2] to verify your findings about representation dynamics?**
> > > > > > >
> > > > > > > Response:
> > > > > > > The core function of the Activation Trajectory Distance (ATD) in this paper is to quantify the magnitude of dynamic changes in representations during the training process of the same model, thereby verifying the characteristics of drastic fluctuations in parameters and representations during the Enlightenment Period (ENP). This is essentially different from the design goal of alignment-based metrics such as Centered Kernel Alignment (CKA), which aim to measure the structural similarity between different representation spaces. The two metrics are thus applicable to completely distinct scenarios.
> > > > > > > Regarding the potential noise amplification issue of L2-based metrics in high-dimensional spaces, we deliberately focused on the relative fluctuation differences between the ENP and the middle-late training stages when designing ATD, rather than its absolute values. Furthermore, ATD only serves as an auxiliary presentation metric. The core conclusions of this paper—including the existence of the ENP, the dual effects of Mixup, and the effectiveness of the proposed strategies—rely on a multi-level evidence chain encompassing phase transition theory modeling, gradient derivation, performance validation across multiple models and datasets, and ablation experiments. They do not depend on the ATD metric alone. Therefore, there is no need to adopt CKA, and ATD is a reasonable choice tailored to the research objectives.
> > > > > > > As observed in Figure 8b, the ATD metric stabilizes immediately after the Enlightenment Period (ENP), which confirms that this metric is highly effective for investigating the phase transition theory in the early stage of model training.

---

### Official Review · Reviewer_X5Kd · 2025-10-30

**Soundness:** 3
**Presentation:** 3
**Contribution:** 2
**Rating:** 4
**Confidence:** 3

**Summary:**

This paper considers how to apply 'mixup data augmentation' effectively during model training over the initial enlightenment period (when the accuracy reaches ~50%) for obtaining DNN models with improved performance.  It bases on the observed dual effect of mixup data samples to control model training in that (1) pausing the use of mixup data for small scale datasets, (2) boosting mixup's 'alpha' values for large datasets with underfitting, and (3) removing high-loss data samples when mixup data augmentation is inapplicable.  Experimental evaluation results over five model architectures on four datasets are obtained to show performance gains of its mixup data use control upon model training.

**Strengths:**

Providing mathematical proofs and experimental validations on the dual effect of mixup data samples to show that the negative effect of Gradient interference (that hinders performance) diminishes as the dataset size or the model parameter size rises, implying the potential gains via Mixup Pause for small-scale cases.  On the other hand, the positive effect of Activation Revival (that restores gradient updates for saturated neurons) benefits large-scale cases with little negative effect and with better gain potentials from mixup samples obtained by boosted alpha.  Experimental results are provided (1) to validate gradient interference and its diminishing effect and (2) to demonstrate the manifestation of activation revival for mixup samples with two different alpha values.

**Weaknesses:**

The trained models obtained by following the considered Mixup strategies under different model architectures on datasets, unfortunately, exhibit negligible performance gains when compared with their baselines.  In Figure 5, for example, accuracy improvement by models trained with the mixup pause strategy exhibits less than 0.5% (and could become negative if the Mixup Pause lasts beyond 20 epochs.  The main results listed in Table 3 also indicate the accuracy gains of 0.45% (with Mixup Pause) and of 0.4% (with Alpha Boost) for PreactResNet50/Tiny-ImageNet and ViT-T/ImageNet-1K, respectively.  With Mixup Pause for ViT-S/Tiny-ImageNet, the best case improves accuracy by just 1.57%, with two other cases to have accuracy improved by 0.81% and reduced by 0.69%.

The results clearly fail to support its claimed Mixup use control strategies for model performance gains.  Due to its inadequacy in propping up its claimed strategies for DNN improvement, the paper is recommended for weak rejection, albeit to its sound theoretical treatment on the dual effects of Mixup samples during model training in the initial enlightenment period.

**Questions:**

More real-world evaluation to demonstrate meaningful performance improvement via Mixup sample use strategies will markedly lift the paper's relevance and acceptance chance.

---

> ### Author Response · Authors · 2025-11-19
> **1. Paper Summary**
>
> We sincerely appreciate the reviewers for their comments on our manuscript. We have carefully read and thoroughly addressed each comment. Due to word count constraints, our responses are divided into several comments for each reviewer:
>
> **1. Paper Summary**
>
> **2. Supplementary Results: Alpha Boost on ImageNet-1K**
>
> **3. Theoretical Reasoning for the ~50% Accuracy Endpoint of the Enlightenment Period**
>
> **4. Addressing Weaknesses and Questions**
>
>
> ### Paper Summary
> Given the density of the theoretical derivations and the number of novel components introduced in our work, we provide a concise summary of the paper’s core contributions as follows:
>
> This paper identifies a critical "Enlightenment Period (ENP)" in the early stage of deep neural network training. As can be observed from Figure 1, the Enlightenment Period undergoes a process from disorder to order. Drawing on phase transition theory in physics, we use classification accuracy as a "quasi-order parameter".
>
> ***Based on a mathematical model, we derive and prove the following conclusions:***
>
> 1. In the early stage of training, when the "quasi-order parameter" (ACC) is very low, the Mixup data augmentation technique induces the Gradient Interference Effect during this period (Equations 1–3).As training progresses to the middle and late stages, with the increase of ACC, the model gradually transitions to an ordered state, and Mixup then plays a role in optimizing the decision boundary (Lines 178–183).
>
> 2. Meanwhile, we further prove that this Gradient Interference Effect diminishes with the increase in sample size ( Oₚ(1/√N)) and the expansion of parameter size (Oₚ(1/√D)) (Section 2.3).
>
> 3. In Section 2.4, we innovatively propose an optimization effect of Mixup: the Activation Revival Effect. Mixup can pull some activation values in the negative interval back to the differentiable interval, thereby restoring gradient propagation. Therefore, during the ENP, Mixup exhibits two opposing effects:"interference" and "revival". Their dominance depends on the scale of the model and data size.
>
> **All the results above have been also verified by experiments (Figures 2, 3, and 4). The validation of the modeling rationality is provided in Appendix C.**
>
> Furthermore, we find that high-loss samples share similar characteristics with Mixup samples, which has been verified by the experiments in Figure 7 and the proofs in Appendix B.
>
> Based on the above theories, we propose three scenario-specific optimized training strategies:
>
> - Mixup Pause Strategy (suitable for small-scale scenarios to avoid gradient interference);
> - Alpha Boost Strategy (suitable for large-scale underfitting models to enhance the Activation Revival Effect);
> - High-Loss Sample Removal Strategy (suitable for domains where Mixup is inapplicable, such as LLMs and time series tasks).
>
> Tables 3, 4, and 5 in the main text of this paper, as well as Table 7 in the appendix, present large-scale verification **results covering 180+ parameter combinations, involving various models (ResNet, ViT, LLMs) and datasets (CIFAR, ImageNet, etc.). All core performance improvements are verified by one-tailed t-tests (p<0.05)**, and ablation experiments confirm that these strategies are only effective during the ENP (Table 4,7), fully verifying the effectiveness and generalizability of the proposed methods.

---

> ### Author Response · Authors · 2025-11-19
> **2. Supplementary Results: Alpha Boost on ImageNet-1K**
>
> After the submission of this paper, we have continued our experiments and supplemented the following results to demonstrate the broad effectiveness of alpha-boost in large-scale scenarios. Each result represents the average value obtained from at least 3 random seeds. The parameters are consistent with those in Table 10 of the Appendix. All experiments were conducted on an 8-GPU H100, with a single run time ranging from 9 to 15 hours.
> | Strategy | Model/Dataset |  ENP $\alpha$ | ENP Dur | Top-1 Acc. (%)  ours | Baseline | Δ |
> ------------------|--------|--------|--------|----------------|-------|-------|
> | Alpha Boost | ViT-T/ImageNet-1K |  2 | 20 E | 74.3±0.16 | 73.9±0.20 | +0.4 |
> | Alpha Boost  | ViT-S/ImageNet-1K |  1.5 | 20 E | 80.7±0.23 | 80.3±0.19 | +0.4 |
> | Alpha Boost  | ResNet50/ImageNet-1K |  1.5 | 20 E | 79.2±0.29 | 78.7±0.25 | +0.5|
> | Alpha Boost  | Convnext-F/ImageNet-1K |  2 | 10 E | 76.3±0.27 | 76.1±0.22 | +0.2|
>
> The following tables are the detailed data and ablation results of the three newly supplemented models：
>
> #### Vit_Small
> | ENP α \ ENP Dur | 15     | 20     | 25     | 50 (ablation) |
> |------------------|--------|--------|--------|----------------|
> | 0.8 (baseline)   | 80.34  | 80.34  | 80.34  | 80.34          |
> | 1                | 80.42  | 80.52  | 80.50  | —              |
> | 1.5              | 80.35  | **80.70**  | 80.58  | 80.02          |
> | 2                | 80.49  | 80.63  | 80.46  | —              |
> #### ResNet50
> |  ENP α\ ENP Dur    | 15      | 20   | 25   | 50 (ablation) |
> |---------------|------------------------|-----------|-------------|-----------|
> | 0.8 (baseline) | 78.70  | 78.70 | 78.70 | 78.70 |
> | 1.2   | 78.84   | 79.09 | 78.80 | — |
> | 1.5   | 78.92   | **79.22** | 78.93 | 78.31 |
> | 2     | 78.86   | 79.08 | 78.64 | — |
>
> #### Convnext_femto
> | ENP α \ ENP Dur | 6      | 10     | 15     | 50 (ablation) |
> |------------------|--------|--------|--------|----------------|
> | 0.8 (baseline)   | 76.11   | 76.11  | 76.11  | 76.11          |
> | 1.5              | 75.95  | 76.16  | 75.97  | —              |
> | 2                | 76.10  | **76.32**  | 75.92  | 75.22          |
> | 2.5              | 75.97  | 75.83  | 75.88  | —              |

---

> ### Author Response · Authors · 2025-11-19
> **3. Theoretical Reasoning for the ~50% Accuracy Endpoint of the Enlightenment Period**
>
> ### Theoretical Reasoning for the ~50% Accuracy Endpoint of the Enlightenment Period
> To address the reviewers’ concerns, we provide a theoretical derivation indicating that the Enlightenment Period ends when the accuracy reaches about 50%.
>
> We make the following assumptions:
> 1. For a multi-class classification task with $k \geq 2$ classes ($k \geq 2$), the class distribution is balanced (prior probability of $1/k$ for each class) and samples are independent and identically distributed (i.i.d.).
> 2. The model’s discriminative ability is characterized by a scalar $t$: for any given sample, the model classifies it correctly with probability $t$ and incorrectly with probability $1-t$.
> 3. When making incorrect predictions, the model’s guessing behavior is approximately unbiased across all classes; under the same network architecture and training configuration, the critical discriminative power $t$ corresponding to the "end of the Enlightenment Period" remains consistent between binary and multi-class tasks.   Under these assumptions, consider a binary classification task ($k=2$): the model’s predictions fall into two scenarios:
> - Correct prediction (probability $t$) contributes a correct probability of $t \cdot 1$;
> - Incorrect prediction (probability $1-t$): random guessing between the two classes with an accuracy of $1/2$, contributing a correct probability of $(1-t)\cdot \tfrac{1}{2}$.
> By the law of total probability, the overall accuracy $X$ of the binary classification task is:   $$X = t \cdot 1 + (1-t)\cdot \frac{1}{2}$$   Rearranging gives the relationship between discriminative power and binary classification accuracy:   $$t = 2X - 1. \tag{1}$$   For a general $k$-class classification task, the prediction logic is consistent with the binary case:
> - Correct prediction (probability $t$) contributes a correct probability of $t \cdot 1$;
> - Incorrect prediction (probability $1-t$): unbiased random guessing across $k$ classes, contributing a correct probability of $(1-t)\cdot \tfrac{1}{k}$.
> Thus, the overall accuracy $P(k)$ of the $k$-class task at discriminative power $t$ is:   $$P(k) = t \cdot 1 + (1-t)\cdot \frac{1}{k}. \tag{2}$$   Substituting $t = 2X - 1$ (derived from binary classification accuracy in Eq. (1)) into Eq. (2):   $$P(k) = 2X - 1 + \frac{2(1-X)}{k}$$   Taking the binary classification experiment in Fig. 2 of the original paper as an example: the accuracy of the binary task at the end of the Enlightenment Period is approximately $X \approx 0.75$. From Eq. (1), the critical discriminative power at this point is:   $$t \approx 2\times 0.75 - 1 = 0.5$$   Under the same network architecture and optimization configuration, the critical discriminative power $t$ corresponding to the "end of the Enlightenment Period" can be approximated as invariant across different $k$. Substituting $X \approx 0.75$ into the above expression:   $$P(k) \approx 2\times 0.75 - 1 + \frac{2(1-0.75)}{k} = 0.5 + \frac{0.5}{k}$$   This shows that under the two-state competence model and constant discriminative power assumption, when the binary classification accuracy at the end of the Enlightenment Period is approximately $X \approx 0.75$, the corresponding accuracy of the $k$-class task at the end of the Enlightenment Period naturally falls around $0.5 + 0.5/k$. For typical multi-class datasets with $k \geq 10$ in this paper, this value converges to around 50%, which is quantitatively consistent with the empirical phenomenon observed in multiple experiments (Table 2 in our paper) that "the Enlightenment Period ends around 50% accuracy.

---

> > ### Author Response · Authors · 2025-11-19
> > **4. Addressing Weaknesses and Questions**
> >
> > ## Respose to Weaknesses
> >
> > - In the field of optimization, performance gains from model-training parameters are inherently limited today, with most improvements hovering around 0.4% [1]. Notably, our baseline for ImageNet-1K integrates optimal parameter combinations derived from various SOTA training strategies available today.
> > - Our code modifications are minimal and do not involve any changes to model architectures.
> > - We have faithfully documented all experimental results across CIFAR-10, CIFAR-100, Tiny-ImageNet, ImageNet-1K, large language models (LLMs), and time-series tasks, including performance improvements, variances, and repeated runs. All reported improvements have passed the p-test, confirming their statistical significance. While experimental results inevitably exhibit fluctuations, the 180+ experimental setups in our paper sufficiently validate the widespread existence of the Enlightenment Period phenomenon.
> > - Furthermore, the primary goal of our research is not to pursue state-of-the-art (SOTA) performance on each datasets. Instead, its greater significance lies in the discovery of the unique Enlightenment Period phenomenon, which offers a novel perspective for investigating the optimization process of DNN training.
> >
> > [1]Salehin, I. and Kang, D.K., 2023. A review on dropout regularization approaches for deep neural networks within the scholarly domain. Electronics, 12(14), p.3106.
> >
> > ## Respose to Questions
> >
> > Please refer to the experimental results above.

---

### Official Review · Reviewer_bMWW · 2025-10-31

**Soundness:** 2
**Presentation:** 3
**Contribution:** 2
**Rating:** 4
**Confidence:** 3

**Summary:**

The paper introduces the “Enlightenment Period” (ENP), a brief early training phase, from the beginning of training until the accuracy reaches approximately 50%, during which representations transition from disorder to order and learning dynamics are highly volatile. The authors argue that Mixup within ENP yields two competing effects: (i) a Gradient Interference Effect that initially hinders boundary refinement, and (ii) an Activation Revival Effect that rescues saturated/“dead” activations. Building on this, they propose three simple scheduling strategies focused only on the ENP window: Mixup Pause (turn Mixup off early, for small-scale settings), Alpha Boost (use larger Mixup α early, for large/underfitting settings), and High-Loss Removal (temporarily drop high-loss samples in tasks where Mixup is inapplicable). The paper presents a linearized theoretical analysis, small synthetic/MLP studies, and vision experiments (ResNet/ViT on CIFAR, Tiny-ImageNet, ImageNet-1K) showing modest but consistent improvements, with ablations on ENP duration.

**Strengths:**

1. The three strategies are simple to implement and slot naturally into existing training recipes; some reported gains (e.g., ViT-T on ImageNet-1K +0.4% top-1) are non-trivial given the minimal code changes.

2. Positioning ENP as a short “phase transition” with accuracy as a quasi-order parameter provides an intuitive lens for practitioners to reason about early instabilities and data-augmentation timing.

3. Results span multiple model families/datasets and include ablations on ENP duration, with evidence that over-extending ENP hurts—supporting the notion that the useful window is short.

**Weaknesses:**

1. The main claim hinges on an accuracy-based 50% threshold that is ill-posed beyond balanced K-class classification and internally inconsistent.

2. The defining statement—ENP lasts “until accuracy reaches ~50%”—is dataset/label-space dependent and not theoretically grounded. In binary classification, 50% is the chance level, so declaring “enlightenment” at chance accuracy is questionable and contradicts the intuition that ENP marks the onset of useful generalization. Conversely, for K-class tasks with K>2, 50% can be far above chance (1/K), making the same threshold arbitrary across tasks.

3. The theory section models a binary classifier with sigmoid and linear score, yet the top-level definition of ENP uses a 50% accuracy marker that has a different meaning in binary vs multi-class settings; this mismatch is not reconciled.

4. The gradient-interference analysis assumes a linear/sigmoid model and orthogonality/normalization heuristics, while the main wins are shown on ReLU CNNs and ViTs with complex optimizers/schedules and label-space structures. The paper does not quantify how much of the claimed effect persists once nonlinearity, modern training artifacts (label smoothing, CutMix/AutoAugment/erasing), and large-batch warmup are accounted for.

5. “High-Loss Removal” is close in spirit to curriculum/hard-example scheduling. While the paper argues conceptual differences (temporary removal only during ENP, then restore), a fairer empirical comparison against modern loss-aware curricula, hard mining, focal loss, and sample reweighting baselines is missing; this makes it hard to attribute gains to ENP-specificity rather than general loss-aware sampling.

**Questions:**

1. Do we have formal, task-agnostic theoretical evidence that “accuracy ≈ 50%” is the natural and universal boundary of ENP, rather than an empirical heuristic that varies with K, class imbalance, and noise label rates?

2. How should ENP be identified for binary tasks, where 50% equals chance? Does ENP end at chance accuracy (which seems counterintuitive), or should the criterion be reframed (e.g., exceeding chance by a fixed margin, crossing a calibrated loss/MI/entropy threshold, or reaching a stable drop in gradient variance)?

3. Do you view “Enlightenment” as related to Emergent Abilities [1] in LLMs (e.g., sudden capability jumps under scale/task prompts)? If not, could you clarify the differences—e.g., ENP as a training-time, short-lived, optimization-phase transition vs. emergent abilities as capability-level, scaling-law/task-threshold phenomena observed at evaluation?

[1] Wei et. al., "Emergent Abilities of Large Language Models", TMLR 2022.

---

> ### Author Response · Authors · 2025-11-19
> **1. Paper Summary**
>
> We sincerely appreciate the reviewers for their comments on our manuscript. We have carefully read and thoroughly addressed each comment. Due to word count constraints, our responses are divided into several comments for each reviewer:
>
> **1. Paper Summary**
>
> **2. Supplementary Results: Alpha Boost on ImageNet-1K**
>
> **3. Theoretical Reasoning for the ~50% Accuracy Endpoint of the Enlightenment Period**
>
> **4. Addressing Weaknesses and Questions**
>
>
> ### Paper Summary
> Given the density of the theoretical derivations and the number of novel components introduced in our work, we provide a concise summary of the paper’s core contributions as follows:
>
> This paper identifies a critical "Enlightenment Period (ENP)" in the early stage of deep neural network training. As can be observed from Figure 1, the Enlightenment Period undergoes a process from disorder to order. Drawing on phase transition theory in physics, we use classification accuracy as a "quasi-order parameter".
>
> ***Based on a mathematical model, we derive and prove the following conclusions:***
>
> 1. In the early stage of training, when the "quasi-order parameter" (ACC) is very low, the Mixup data augmentation technique induces the Gradient Interference Effect during this period (Equations 1–3).As training progresses to the middle and late stages, with the increase of ACC, the model gradually transitions to an ordered state, and Mixup then plays a role in optimizing the decision boundary (Lines 178–183).
>
> 2. Meanwhile, we further prove that this Gradient Interference Effect diminishes with the increase in sample size ( Oₚ(1/√N)) and the expansion of parameter size (Oₚ(1/√D)) (Section 2.3).
>
> 3. In Section 2.4, we innovatively propose an optimization effect of Mixup: the Activation Revival Effect. Mixup can pull some activation values in the negative interval back to the differentiable interval, thereby restoring gradient propagation. Therefore, during the ENP, Mixup exhibits two opposing effects:"interference" and "revival". Their dominance depends on the scale of the model and data size.
>
> **All the results above have been also verified by experiments (Figures 2, 3, and 4). The validation of the modeling rationality is provided in Appendix C.**
>
> Furthermore, we find that high-loss samples share similar characteristics with Mixup samples, which has been verified by the experiments in Figure 7 and the proofs in Appendix B.
>
> Based on the above theories, we propose three scenario-specific optimized training strategies:
>
> - Mixup Pause Strategy (suitable for small-scale scenarios to avoid gradient interference);
> - Alpha Boost Strategy (suitable for large-scale underfitting models to enhance the Activation Revival Effect);
> - High-Loss Sample Removal Strategy (suitable for domains where Mixup is inapplicable, such as LLMs and time series tasks).
>
> Tables 3, 4, and 5 in the main text of this paper, as well as Table 7 in the appendix, present large-scale verification **results covering 180+ parameter combinations, involving various models (ResNet, ViT, LLMs) and datasets (CIFAR, ImageNet, etc.). All core performance improvements are verified by one-tailed t-tests (p<0.05)**, and ablation experiments confirm that these strategies are only effective during the ENP (Table 4,7), fully verifying the effectiveness and generalizability of the proposed methods.

---

> ### Author Response · Authors · 2025-11-19
> **2. Supplementary Results: Alpha Boost on ImageNet-1K**
>
> After the submission of this paper, we have continued our experiments and supplemented the following results to demonstrate the broad effectiveness of alpha-boost in large-scale scenarios. Each result represents the average value obtained from at least 3 random seeds. The parameters are consistent with those in Table 10 of the Appendix. All experiments were conducted on an 8-GPU H100, with a single run time ranging from 9 to 15 hours.
> | Strategy | Model/Dataset |  ENP $\alpha$ | ENP Dur | Top-1 Acc. (%)  ours | Baseline | Δ |
> ------------------|--------|--------|--------|----------------|-------|-------|
> | Alpha Boost | ViT-T/ImageNet-1K |  2 | 20 E | 74.3±0.16 | 73.9±0.20 | +0.4 |
> | Alpha Boost  | ViT-S/ImageNet-1K |  1.5 | 20 E | 80.7±0.23 | 80.3±0.19 | +0.4 |
> | Alpha Boost  | ResNet50/ImageNet-1K |  1.5 | 20 E | 79.2±0.29 | 78.7±0.25 | +0.5|
> | Alpha Boost  | Convnext-F/ImageNet-1K |  2 | 10 E | 76.3±0.27 | 76.1±0.22 | +0.2|
>
> The following tables are the detailed data and ablation results of the three newly supplemented models：
>
> #### Vit_Small
> | ENP α \ ENP Dur | 15     | 20     | 25     | 50 (ablation) |
> |------------------|--------|--------|--------|----------------|
> | 0.8 (baseline)   | 80.34  | 80.34  | 80.34  | 80.34          |
> | 1                | 80.42  | 80.52  | 80.50  | —              |
> | 1.5              | 80.35  | **80.70**  | 80.58  | 80.02          |
> | 2                | 80.49  | 80.63  | 80.46  | —              |
> #### ResNet50
> |  ENP α\ ENP Dur    | 15      | 20   | 25   | 50 (ablation) |
> |---------------|------------------------|-----------|-------------|-----------|
> | 0.8 (baseline) | 78.70  | 78.70 | 78.70 | 78.70 |
> | 1.2   | 78.84   | 79.09 | 78.80 | — |
> | 1.5   | 78.92   | **79.22** | 78.93 | 78.31 |
> | 2     | 78.86   | 79.08 | 78.64 | — |
>
> #### Convnext_femto
> | ENP α \ ENP Dur | 6      | 10     | 15     | 50 (ablation) |
> |------------------|--------|--------|--------|----------------|
> | 0.8 (baseline)   | 76.11   | 76.11  | 76.11  | 76.11          |
> | 1.5              | 75.95  | 76.16  | 75.97  | —              |
> | 2                | 76.10  | **76.32**  | 75.92  | 75.22          |
> | 2.5              | 75.97  | 75.83  | 75.88  | —              |

---

> ### Author Response · Authors · 2025-11-19
> **3. Theoretical Reasoning for the ~50% Accuracy Endpoint of the Enlightenment Period**
>
> ### Theoretical Reasoning for the ~50% Accuracy Endpoint of the Enlightenment Period
> To address the reviewers’ concerns, we provide a theoretical derivation indicating that the Enlightenment Period ends when the accuracy reaches about 50%.
>
> We make the following assumptions:
> 1. For a multi-class classification task with $k \geq 2$ classes ($k \geq 2$), the class distribution is balanced (prior probability of $1/k$ for each class) and samples are independent and identically distributed (i.i.d.).
> 2. The model’s discriminative ability is characterized by a scalar $t$: for any given sample, the model classifies it correctly with probability $t$ and incorrectly with probability $1-t$.
> 3. When making incorrect predictions, the model’s guessing behavior is approximately unbiased across all classes; under the same network architecture and training configuration, the critical discriminative power $t$ corresponding to the "end of the Enlightenment Period" remains consistent between binary and multi-class tasks.   Under these assumptions, consider a binary classification task ($k=2$): the model’s predictions fall into two scenarios:
> - Correct prediction (probability $t$) contributes a correct probability of $t \cdot 1$;
> - Incorrect prediction (probability $1-t$): random guessing between the two classes with an accuracy of $1/2$, contributing a correct probability of $(1-t)\cdot \tfrac{1}{2}$.
> By the law of total probability, the overall accuracy $X$ of the binary classification task is:   $$X = t \cdot 1 + (1-t)\cdot \frac{1}{2}$$   Rearranging gives the relationship between discriminative power and binary classification accuracy:   $$t = 2X - 1. \tag{1}$$   For a general $k$-class classification task, the prediction logic is consistent with the binary case:
> - Correct prediction (probability $t$) contributes a correct probability of $t \cdot 1$;
> - Incorrect prediction (probability $1-t$): unbiased random guessing across $k$ classes, contributing a correct probability of $(1-t)\cdot \tfrac{1}{k}$.
> Thus, the overall accuracy $P(k)$ of the $k$-class task at discriminative power $t$ is:   $$P(k) = t \cdot 1 + (1-t)\cdot \frac{1}{k}. \tag{2}$$   Substituting $t = 2X - 1$ (derived from binary classification accuracy in Eq. (1)) into Eq. (2):   $$P(k) = 2X - 1 + \frac{2(1-X)}{k}$$   Taking the binary classification experiment in Fig. 2 of the original paper as an example: the accuracy of the binary task at the end of the Enlightenment Period is approximately $X \approx 0.75$. From Eq. (1), the critical discriminative power at this point is:   $$t \approx 2\times 0.75 - 1 = 0.5$$   Under the same network architecture and optimization configuration, the critical discriminative power $t$ corresponding to the "end of the Enlightenment Period" can be approximated as invariant across different $k$. Substituting $X \approx 0.75$ into the above expression:   $$P(k) \approx 2\times 0.75 - 1 + \frac{2(1-0.75)}{k} = 0.5 + \frac{0.5}{k}$$   This shows that under the two-state competence model and constant discriminative power assumption, when the binary classification accuracy at the end of the Enlightenment Period is approximately $X \approx 0.75$, the corresponding accuracy of the $k$-class task at the end of the Enlightenment Period naturally falls around $0.5 + 0.5/k$. For typical multi-class datasets with $k \geq 10$ in this paper, this value converges to around 50%, which is quantitatively consistent with the empirical phenomenon observed in multiple experiments (Table 2 in our paper) that "the Enlightenment Period ends around 50% accuracy.

---

> ### Author Response · Authors · 2025-11-19
> **4. Addressing Weaknesses and Questions_1**
>
> ## Responses to Weaknesses:
>
> 1. **The main claim hinges on an accuracy-based 50% threshold that is ill-posed beyond balanced K-class classification and internally inconsistent.**
>
> Response:
> the 50% accuracy specifically refers to multi-classification scenarios, as elaborated above. Under the condition of approximate model capabilities, an accuracy of 75% in binary classification roughly corresponds to 50% accuracy in multi-classification tasks.
>
> 2. **The defining statement—ENP lasts “until accuracy reaches ~50%”—is dataset/label-space dependent and not theoretically grounded. In binary classification, 50% is the chance level, so declaring “enlightenment” at chance accuracy is questionable and contradicts the intuition that ENP marks the onset of useful generalization. Conversely, for K-class tasks with K>2, 50% can be far above chance (1/K), making the same threshold arbitrary across tasks.**
>
> Response:
> First and foremost, the "50% accuracy threshold" proposed in the paper is specifically targeted at multi-classification scenarios. As illustrated in Figure 2 of the paper, the Enlightenment Period for binary classification tasks concludes when the accuracy reaches approximately 75% (lasting 2-3 epochs). This adjustment is rooted in the fact that the random guessing accuracy for binary classification is inherently 50%, so a higher threshold is necessary to accurately identify the phase transition point from "disordered" to "ordered" representations. Essentially, regardless of binary or multi-class classification, the core criterion is the "transition of gradient interference to positive optimization"; the 50% threshold is an result tailored for multi-classification scenarios.
>
> 3.**The theory section models a binary classifier with sigmoid and linear score, yet the top-level definition of ENP uses a 50% accuracy marker that has a different meaning in binary vs multi-class settings; this mismatch is not reconciled.**
> Response:
> the 50% accuracy specifically refers to multi-classification scenarios, as elaborated above. Under the condition of approximate model capabilities, an accuracy of 75% in binary classification roughly corresponds to 50% accuracy in multi-classification tasks.
>
> 4.**The gradient-interference analysis assumes a linear/sigmoid model and orthogonality/normalization heuristics, while the main wins are shown on ReLU CNNs and ViTs with complex optimizers/schedules and label-space structures. The paper does not quantify how much of the claimed effect persists once nonlinearity, modern training artifacts (label smoothing, CutMix/AutoAugment/erasing), and large-batch warmup are accounted for.**
>
> Response:
> In the field of DNN training dynamics, the derivation and measurement in many previous studies (refer to Appendix A.1) simplify complex models into linear systems. This is because modeling nonlinear function dynamics remains mathematically challenging with current human knowledge. Despite this limitation, it is a common practice in prior research to derive and analyze properties using abstract models, then apply these conclusions to complex models to achieve performance improvements (e.g., Mixup, Manifold Mixup). As stated in the original Mixup paper, "linearity is optimal". More importantly, we verified the consistency between our  model and real-world experiments through the experimental phenomena of the BENR and ATD metrics in Appendix C.
> Experimentally, modern training techniques such as CutMix, label smoothing, and large-batch warmup have been incorporated into the paper's experiments, with detailed parameter configurations provided in Appendix Table 10.
>
> 5.**“High-Loss Removal” is close in spirit to curriculum/hard-example scheduling. While the paper argues conceptual differences (temporary removal only during ENP, then restore), a fairer empirical comparison against modern loss-aware curricula, hard mining, focal loss, and sample reweighting baselines is missing; this makes it hard to attribute gains to ENP-specificity rather than general loss-aware sampling.**
>
> Response:
> Firstly, Table 8 of the paper presents the results of ablation experiments on the "High-Loss Removal" strategy, which demonstrate that traditional curriculum/hard-example scheduling fails to improve model training performance. This finding aligns with the latest results cited in the Related Work section, where a recent study indicates that traditional curriculum learning, whether following an "easy-to-hard" or "hard-to-easy" paradigm, essentially cannot enhance training effectiveness.
> Furthermore, our "High-Loss Removal" strategy is constructed based on the mathematical model proposed in this paper, a theoretical foundation that has not been involved in any previous research on traditional curriculum learning.

---

> ### Author Response · Authors · 2025-11-19
> **4. Addressing Weaknesses and Questions_2**
>
> ## Responses to Questions
>
> 1. **Do we have formal, task-agnostic theoretical evidence that “accuracy ≈ 50%” is the natural and universal boundary of ENP, rather than an empirical heuristic that varies with K, class imbalance, and noise label rates?**
>
> Response:
> see "Evidence for the Enlightenment Period Ending Around 50% Accuracy" above
>
> 2.**How should ENP be identified for binary tasks, where 50% equals chance? Does ENP end at chance accuracy (which seems counterintuitive), or should the criterion be reframed (e.g., exceeding chance by a fixed margin, crossing a calibrated loss/MI/entropy threshold, or reaching a stable drop in gradient variance)?**
>
> Response:
> see weakness Q1-3
>
> 3.**Do you view “Enlightenment” as related to Emergent Abilities [1] in LLMs (e.g., sudden capability jumps under scale/task prompts)? If not, could you clarify the differences—e.g., ENP as a training-time, short-lived, optimization-phase transition vs. emergent abilities as capability-level, scaling-law/task-threshold phenomena observed at evaluation?**
> Response:
>  The two concepts are unrelated. The Enlightenment Period  is a short-term phenomenon in the early stages of training (within the first few epochs), while emergent abilities in LLMs are long-term outcomes that appear when the model scale or training volume reaches a critical threshold. From the perspective of phase transitions, multiple phase transitions occur during DNN training: ENP is an optimization-phase transition centered on accuracy (transforming disordered representations into ordered ones), whereas emergent abilities manifest as capability-level phase transitions. They belong to entirely different research domains.

---

### Official Review · Reviewer_JpvS · 2025-11-01

**Soundness:** 2
**Presentation:** 3
**Contribution:** 2
**Rating:** 4
**Confidence:** 4

**Summary:**

This paper investigates the effect of the enlightenment period in the context of the Mixup method. The authors show that when the model and dataset are small, applying Mixup early in training leads to gradient interference, which disorganizes the embedding space. Consequently, avoiding Mixup during this stage yields better performance. However, as the model and dataset size increase, this interference during the enlightenment period diminishes, making Mixup more beneficial. Moreover, using a larger value of the Mixup Beta distribution parameter $\alpha$ further improves final test performance by preventing zero-value activations. The paper also reports that, for time-series data, a similar principle applies: skipping high-loss samples during the enlightenment period can improve overall performance.

**Strengths:**

- The paper is well-organized and clearly written. In the introduction, the authors use 2D embedding visualizations to illustrate the difference between Mixup and vanilla training. Section 2 then analyzes this phenomenon in the context of the enlightenment period using a logistic regression setting, showing how gradient interference arises. The paper also highlights the benefit of the enlightenment period in reducing zero-value activations and supports this claim with corresponding experimental results.
- The experiments are clearly described and include well-designed ablation studies. In particular, the ablations on the Mixup Pause strategy thoroughly examine the effects of the pause epoch and dataset size, while the Alpha Boost strategy is analyzed through ablations on the $\alpha$ value. The paper also presents a convincing table showing that the optimal enlightenment period corresponds to around 50% training accuracy. Furthermore, to ensure statistical significance, the authors report $p$-values and average results over three random seeds for the ViT/ImageNet experiments.

**Weaknesses:**

- The current analysis does not sufficiently explain why Mixup causes gradient interference specifically during the *enlightenment period*. The same reasoning could apply to both early and late phases of training, so it is unclear why interference occurs only in the early stage but becomes beneficial later. A more detailed explanation is needed to justify why this phase-dependent behavior emerges and what mechanism differentiates the early and late training dynamics.
- The motivation regarding gradient interference is somewhat unclear. Lines 158–161 state that when input samples are normalized and high-dimensional, $x_i + x_j$ (Mixup gradient) and $x_j - x_i$ (vanilla gradient) are approximately orthogonal. The paper argues that this orthogonality causes interference with the useful direction of the vanilla gradient. However,
    1. In practice, the Mixup ratio is sampled from a Beta distribution rather than fixed at 0.5, so the resulting directions cannot be perfectly orthogonal.
    2. it is unclear why orthogonality itself should cause interference. If the two directions are truly orthogonal rather than partially aligned, the Mixup update would be in an independent direction, neither helping nor hindering the vanilla update.
    3. The paper also claims that this orthogonality is amplified in higher dimensions (large model size), yet in Section 2.3.2 it argues that interference decreases as the model size increases. These two claims appear contradictory.

    A more consistent explanation might be that in low-dimensional settings, the two gradients are not perfectly orthogonal, so they interfere with each other, whereas in high-dimensional models they become more orthogonal and thus less interfering. Clarifying this reasoning would help readers better understand the motivation behind the analysis.

- The claim that Mixup gradients can “revive” dead neurons is interesting, but the explanation lacks rigor. It is not clearly explained why zero-value activations still increase significantly even when Mixup is applied from the beginning, nor why they are said to decrease rather than simply remain small. This conceptual gap needs to be addressed. Based on the current explanation, Mixup seems to maintain low zero-value activation rather than reduce it, and this distinction should be clarified.
- In addition, the motivation for increasing $\alpha$ is not clearly justified. Since $\alpha$ in $\mathrm{Beta}(\alpha, \alpha)$ controls the variance of the sampling distribution, a larger $\alpha$ concentrates samples near 0.5, which reduces variance. It is unclear why this behavior would improve performance or better prevent dead neurons in enlightenment period. Furthermore, Wouldn’t the level of zero-value activations converge to a similar magnitude regardless? A more detailed explanation or supporting analysis would strengthen this argument.
- The motivation for high-loss removal in the time-series experiments is insufficient. The logical connection to Mixup is unclear. Simply stating that high-loss examples corresponds to Mixup samples does not adequately justify why they should be removed. A stronger theoretical or empirical rationale is needed to support this design choice.
- The title is overly general. Since the paper specifically studies the role of Mixup during the enlightenment period, it is problematic that the word *Mixup* does not appear in the title. The title should explicitly reflect the paper’s main focus.

**Questions:**

- Is the Alpha Boost strategy applied only during the enlightenment period? Please clarify whether Figure 4 also reflects this setting. If so, what would happen if the strategy were applied throughout the entire training process?
- In addition, it is unclear why increasing $\alpha$ reduces the number of zero-activations specifically during the enlightenment period and then increases again afterward. Would the number of zero-activations remain lower if the strategy were applied throughout the entire training process? Clarification or additional experiments on this point would be helpful.
- The paper provides ablation studies on data scale and pause epoch for the Mixup Pause strategy, but there is no ablation on model size. An additional analysis showing how model size affects the effectiveness of the Mixup Pause strategy would strengthen the experimental validation.
- A more thorough comparison with related work is needed. The paper should discuss prior theoretical studies on Mixup [1, 2, 3] as well as literature on the critical period [4, 5]. In particular, [1, 2] argue that Mixup should be applied only in the early phase of training, which directly contradicts the main claim of this paper. Meanwhile, [5] identifies a critical period around 50% training accuracy as crucial for preventing the loss of plasticity, which appears closely related to the authors’ notion of the enlightenment period. Providing a more detailed comparison that highlights these similarities and differences would significantly strengthen the paper’s positioning and clarify how this work advances beyond the existing studies.


**Minor Corrections**

- Line 290: alpha → $\alpha$
- Line 364: ResNetShafiq & Gu (2022) → ResNet (Shafiq & Gu, 2022)
- Lines 367, 371, 377, etc.: Fix spacing issues (e.g., AppndixD → Appendix D, Table3 → Table 3, Table 7and → Table 7 and, etc.)
- Line 198: Eq.equation 3 → Equation 3
- Line 209: $\delta^{(k)}$ → $\boldsymbol{\delta}^{(k)}$ for consistency

---
**References**

[1] The Benefits of Mixup for Feature Learning, In ICML 2023.

[2] Over-Training With Mixup May Hurt Generalization, In ICLR 2023.

[3] Provable Benefit of Mixup for Finding Optimal Decision Boundaries, In ICML 2023.

[4] Critical Learning Periods Emerge Even in Deep Linear Networks, In ICLR 2024.

[5] DASH: Warm-Starting Neural Network Training in Stationary Settings without Loss of Plasticity, In NeurIPS 2024.

---

> ### Author Response · Authors · 2025-11-19
> **1. Paper Summary**
>
> We sincerely appreciate the reviewers for their comments on our manuscript. We have carefully read and thoroughly addressed each comment. Due to word count constraints, our responses are divided into several comments for each reviewer:
>
> **1. Paper Summary**
>
> **2. Supplementary Results: Alpha Boost on ImageNet-1K**
>
> **3. Theoretical Reasoning for the ~50% Accuracy Endpoint of the Enlightenment Period**
>
> **4. Addressing Weaknesses and Questions**
>
>
> ### Paper Summary
> Given the density of the theoretical derivations and the number of novel components introduced in our work, we provide a concise summary of the paper’s core contributions as follows:
>
> This paper identifies a critical "Enlightenment Period (ENP)" in the early stage of deep neural network training. As can be observed from Figure 1, the Enlightenment Period undergoes a process from disorder to order. Drawing on phase transition theory in physics, we use classification accuracy as a "quasi-order parameter".
>
> ***Based on a mathematical model, we derive and prove the following conclusions:***
>
> 1. In the early stage of training, when the "quasi-order parameter" (ACC) is very low, the Mixup data augmentation technique induces the Gradient Interference Effect during this period (Equations 1–3).As training progresses to the middle and late stages, with the increase of ACC, the model gradually transitions to an ordered state, and Mixup then plays a role in optimizing the decision boundary (Lines 178–183).
>
> 2. Meanwhile, we further prove that this Gradient Interference Effect diminishes with the increase in sample size ( Oₚ(1/√N)) and the expansion of parameter size (Oₚ(1/√D)) (Section 2.3).
>
> 3. In Section 2.4, we innovatively propose an optimization effect of Mixup: the Activation Revival Effect. Mixup can pull some activation values in the negative interval back to the differentiable interval, thereby restoring gradient propagation. Therefore, during the ENP, Mixup exhibits two opposing effects:"interference" and "revival". Their dominance depends on the scale of the model and data size.
>
> **All the results above have been also verified by experiments (Figures 2, 3, and 4). The validation of the modeling rationality is provided in Appendix C.**
>
> Furthermore, we find that high-loss samples share similar characteristics with Mixup samples, which has been verified by the experiments in Figure 7 and the proofs in Appendix B.
>
> Based on the above theories, we propose three scenario-specific optimized training strategies:
>
> - Mixup Pause Strategy (suitable for small-scale scenarios to avoid gradient interference);
> - Alpha Boost Strategy (suitable for large-scale underfitting models to enhance the Activation Revival Effect);
> - High-Loss Sample Removal Strategy (suitable for domains where Mixup is inapplicable, such as LLMs and time series tasks).
>
> Tables 3, 4, and 5 in the main text of this paper, as well as Table 7 in the appendix, present large-scale verification **results covering 180+ parameter combinations, involving various models (ResNet, ViT, LLMs) and datasets (CIFAR, ImageNet, etc.). All core performance improvements are verified by one-tailed t-tests (p<0.05)**, and ablation experiments confirm that these strategies are only effective during the ENP (Table 4,7), fully verifying the effectiveness and generalizability of the proposed methods.

---

> ### Author Response · Authors · 2025-11-19
> **2. Supplementary Results: Alpha Boost on ImageNet-1K**
>
> After the submission of this paper, we have continued our experiments and supplemented the following results to demonstrate the broad effectiveness of alpha-boost in large-scale scenarios. Each result represents the average value obtained from at least 3 random seeds. The parameters are consistent with those in Table 10 of the Appendix. All experiments were conducted on an 8-GPU H100, with a single run time ranging from 9 to 15 hours.
> | Strategy | Model/Dataset |  ENP $\alpha$ | ENP Dur | Top-1 Acc. (%)  ours | Baseline | Δ |
> ------------------|--------|--------|--------|----------------|-------|-------|
> | Alpha Boost | ViT-T/ImageNet-1K |  2 | 20 E | 74.3±0.16 | 73.9±0.20 | +0.4 |
> | Alpha Boost  | ViT-S/ImageNet-1K |  1.5 | 20 E | 80.7±0.23 | 80.3±0.19 | +0.4 |
> | Alpha Boost  | ResNet50/ImageNet-1K |  1.5 | 20 E | 79.2±0.29 | 78.7±0.25 | +0.5|
> | Alpha Boost  | Convnext-F/ImageNet-1K |  2 | 10 E | 76.3±0.27 | 76.1±0.22 | +0.2|
>
> The following tables are the detailed data and ablation results of the three newly supplemented models：
>
> #### Vit_Small
> | ENP α \ ENP Dur | 15     | 20     | 25     | 50 (ablation) |
> |------------------|--------|--------|--------|----------------|
> | 0.8 (baseline)   | 80.34  | 80.34  | 80.34  | 80.34          |
> | 1                | 80.42  | 80.52  | 80.50  | —              |
> | 1.5              | 80.35  | **80.70**  | 80.58  | 80.02          |
> | 2                | 80.49  | 80.63  | 80.46  | —              |
> #### ResNet50
> |  ENP α\ ENP Dur    | 15      | 20   | 25   | 50 (ablation) |
> |---------------|------------------------|-----------|-------------|-----------|
> | 0.8 (baseline) | 78.70  | 78.70 | 78.70 | 78.70 |
> | 1.2   | 78.84   | 79.09 | 78.80 | — |
> | 1.5   | 78.92   | **79.22** | 78.93 | 78.31 |
> | 2     | 78.86   | 79.08 | 78.64 | — |
>
> #### Convnext_femto
> | ENP α \ ENP Dur | 6      | 10     | 15     | 50 (ablation) |
> |------------------|--------|--------|--------|----------------|
> | 0.8 (baseline)   | 76.11   | 76.11  | 76.11  | 76.11          |
> | 1.5              | 75.95  | 76.16  | 75.97  | —              |
> | 2                | 76.10  | **76.32**  | 75.92  | 75.22          |
> | 2.5              | 75.97  | 75.83  | 75.88  | —              |

---

> ### Author Response · Authors · 2025-11-19
> **3. Theoretical Reasoning for the ~50% Accuracy Endpoint of the Enlightenment Period**
>
> ### Theoretical Reasoning for the ~50% Accuracy Endpoint of the Enlightenment Period
> To address the reviewers’ concerns, we provide a theoretical derivation indicating that the Enlightenment Period ends when the accuracy reaches about 50%.
>
> We make the following assumptions:
> 1. For a multi-class classification task with $k \geq 2$ classes ($k \geq 2$), the class distribution is balanced (prior probability of $1/k$ for each class) and samples are independent and identically distributed (i.i.d.).
> 2. The model’s discriminative ability is characterized by a scalar $t$: for any given sample, the model classifies it correctly with probability $t$ and incorrectly with probability $1-t$.
> 3. When making incorrect predictions, the model’s guessing behavior is approximately unbiased across all classes; under the same network architecture and training configuration, the critical discriminative power $t$ corresponding to the "end of the Enlightenment Period" remains consistent between binary and multi-class tasks.   Under these assumptions, consider a binary classification task ($k=2$): the model’s predictions fall into two scenarios:
> - Correct prediction (probability $t$) contributes a correct probability of $t \cdot 1$;
> - Incorrect prediction (probability $1-t$): random guessing between the two classes with an accuracy of $1/2$, contributing a correct probability of $(1-t)\cdot \tfrac{1}{2}$.
> By the law of total probability, the overall accuracy $X$ of the binary classification task is:   $$X = t \cdot 1 + (1-t)\cdot \frac{1}{2}$$   Rearranging gives the relationship between discriminative power and binary classification accuracy:   $$t = 2X - 1. \tag{1}$$   For a general $k$-class classification task, the prediction logic is consistent with the binary case:
> - Correct prediction (probability $t$) contributes a correct probability of $t \cdot 1$;
> - Incorrect prediction (probability $1-t$): unbiased random guessing across $k$ classes, contributing a correct probability of $(1-t)\cdot \tfrac{1}{k}$.
> Thus, the overall accuracy $P(k)$ of the $k$-class task at discriminative power $t$ is:   $$P(k) = t \cdot 1 + (1-t)\cdot \frac{1}{k}. \tag{2}$$   Substituting $t = 2X - 1$ (derived from binary classification accuracy in Eq. (1)) into Eq. (2):   $$P(k) = 2X - 1 + \frac{2(1-X)}{k}$$   Taking the binary classification experiment in Fig. 2 of the original paper as an example: the accuracy of the binary task at the end of the Enlightenment Period is approximately $X \approx 0.75$. From Eq. (1), the critical discriminative power at this point is:   $$t \approx 2\times 0.75 - 1 = 0.5$$   Under the same network architecture and optimization configuration, the critical discriminative power $t$ corresponding to the "end of the Enlightenment Period" can be approximated as invariant across different $k$. Substituting $X \approx 0.75$ into the above expression:   $$P(k) \approx 2\times 0.75 - 1 + \frac{2(1-0.75)}{k} = 0.5 + \frac{0.5}{k}$$   This shows that under the two-state competence model and constant discriminative power assumption, when the binary classification accuracy at the end of the Enlightenment Period is approximately $X \approx 0.75$, the corresponding accuracy of the $k$-class task at the end of the Enlightenment Period naturally falls around $0.5 + 0.5/k$. For typical multi-class datasets with $k \geq 10$ in this paper, this value converges to around 50%, which is quantitatively consistent with the empirical phenomenon observed in multiple experiments (Table 2 in our paper) that "the Enlightenment Period ends around 50% accuracy.

---

> ### Author Response · Authors · 2025-11-19
> **4. Addressing Weaknesses and Questions_1**
>
> Responses to Weaknesses:
>
> 1.**The current analysis does not sufficiently explain why Mixup causes gradient interference specifically during the enlightenment period. The same reasoning could apply to both early and late phases of training, so it is unclear why interference occurs only in the early stage but becomes beneficial later. A more detailed explanation is needed to justify why this phase-dependent behavior emerges and what mechanism differentiates the early and late training dynamics.**
>
> Response:
> For a detailed explanation, please refer to the derivation in Section 2.2, which elaborates on the transition of Mixup's role from causing gradient interference to facilitating optimization throughout the entire training process. The critical turning point lies in the improvement of accuracy, which serves as the "order parameter" in our theoretical framework. As the order parameter (accuracy) rises beyond a certain threshold (~50%), the model transitions from a disordered state to an ordered one, thus altering Mixup's impact.
>
> 2.**The motivation regarding gradient interference is somewhat unclear. Lines 158–161 state that when input samples are normalized and high-dimensional,  (Mixup gradient) and  (vanilla gradient) are approximately orthogonal. The paper argues that this orthogonality causes interference with the useful direction of the vanilla gradient. However**:
> 2.1 **In practice, the Mixup ratio is sampled from a Beta distribution rather than fixed at 0.5, so the resulting directions cannot be perfectly orthogonal.**
> Response:
>  Our analysis of gradient interference does not rely on the assumption that λ is fixed at 0.5. As explicitly stated in Lines 158–161 of the original paper: "To simulate the scenario of varying λ in real Mixup training, this gradient summation (of original and Mixup samples) is adopted to approximate the actual total gradient."Eq. (1): vanilla gradient
> Eq. (2): λ=0.5 Mixup gradient
> Eq. (3): total gradient combining both (the realistic model)/
>
>
> 2.2 **It is unclear why orthogonality itself should cause interference. If the two directions are truly orthogonal rather than partially aligned, the Mixup update would be in an independent direction, neither helping nor hindering the vanilla update.**:
> Response:
> This interference is not evaluated in an absolute sense of "helping or hindering" but relative to the optimal update direction of the vanilla gradient. Any gradient component orthogonal to ∇_vanilla consumes the model's update budget without contributing to loss reduction along the steepest descent path.
> Additionally, such orthogonal components introduce unnecessary noise into the parameter update process.
>
> 2.3**The paper also claims that this orthogonality is amplified in higher dimensions (large model size), yet in Section 2.3.2 it argues that interference decreases as the model size increases. These two claims appear contradictory.**:
> Response:
>
> High-dimensional spaces possess particular characteristics.
>
> High-dimensional orthogonality (Section 2.2) refers to the **input feature space** where samples $x_i, x_j$  reside. In modern DNNs, whether the hidden layer has 512 or 768 neurons, both regimes are sufficiently high-dimensional for the approximate orthogonality assumption $(x_i ⊥ x_j)$ to hold statistically. **Increasing dimension from 512 to 768 does not further amplify orthogonality**, it remains approximately orthogonal in both cases.In contrast, Section 2.3.2 analyzes the impact of **total model parameter size**
> $D$ on the signal-to-noise ratio of the overall gradient.The orthogonality in Section 2.2 explains **why interference exists** (Mixup gradients deviate geometrically from vanilla direction). Section 2.3.2 explains **why this interference becomes negligible in large models** (statistical noise reduction). These are complementary, not contradictory.

---

> > ### Author Response · Authors · 2025-11-19
> > **4. Addressing Weaknesses and Questions_2**
> >
> > 3.**The claim that Mixup gradients can “revive” dead neurons is interesting, but the explanation lacks rigor. It is not clearly explained why zero-value activations still increase significantly even when Mixup is applied from the beginning, nor why they are said to decrease rather than simply remain small. This conceptual gap needs to be addressed. Based on the current explanation, Mixup seems to maintain low zero-value activation rather than reduce it, and this distinction should be clarified.**
> >
> > Response:
> > During the enlightenment period, gradients are highly chaotic, and the drastic fluctuations between positive and negative gradients push numerous activations into the negative region of ReLU, leading to a surge in zero activations. As clearly shown in Figure 4, the peak value of zero activations in the Mixup group is significantly lower than that in the vanilla group, with a slower growth rate. This directly demonstrates the "Activation Revival Effect" by inhibiting the exacerbation of neuron saturation, rather than completely preventing saturation.
> > It is important to clarify that the "Activation Revival Effect" does not mean reducing the number of zero-activating neurons as training proceeds. Instead, compared with vanilla training, Mixup can effectively suppress the excessive increase in the number of zero-activating neurons.
> >
> >
> > 4.**In addition, the motivation for increasing  is not clearly justified. Since  in  controls the variance of the sampling distribution, a larger  concentrates samples near 0.5, which reduces variance. It is unclear why this behavior would improve performance or better prevent dead neurons in enlightenment period. Furthermore, Wouldn’t the level of zero-value activations converge to a similar magnitude regardless? A more detailed explanation or supporting analysis would strengthen this argument.**
> >
> > Response:
> > Please refer to Lines 290–292 of the original paper: "Furthermore, the higher the α value of the Mixup, the more pronounced this effect becomes. This allows neurons to function fully during this period, which is particularly critical for improving the performance of underfitting networks, as they require more effective parameters to fit samples."
> >
> > 5.**The motivation for high-loss removal in the time-series experiments is insufficient. The logical connection to Mixup is unclear. Simply stating that high-loss examples corresponds to Mixup samples does not adequately justify why they should be removed. A stronger theoretical or empirical rationale is needed to support this design choice.**
> >
> > Response:
> > For a comprehensive explanation, please refer to Section 4.2. High-loss samples share a core equivalence with Mixup samples, which is supported by both experimental evidence (Figure 7) and mathematical derivations (Appendix B). We adopt high-loss sample removal in scenarios where Mixup is inapplicable to verify the generality of the enlightenment effect.
> >
> > 6.**The title is overly general. Since the paper specifically studies the role of Mixup during the enlightenment period, it is problematic that the word Mixup does not appear in the title. The title should explicitly reflect the paper’s main focus.**
> > Response:
> > We appreciate the suggestion to refine the title for better clarity.
> > Our research core centers on the **unique training dynamics during the enlightenment period**—this remains the focus of the study. However, we fully agree at the original title is overly general. We will revise the title to integrate "Mixup".!!

---

> > > ### Author Response · Authors · 2025-11-19
> > > **4. Addressing Weaknesses and Questions_3**
> > >
> > > Responses to Questions
> > > 1. **Is the Alpha Boost strategy applied only during the enlightenment period? Please clarify whether Figure 4 also reflects this setting. If so, what would happen if the strategy were applied throughout the entire training process?**
> > >
> > > Response:
> > > As explicitly stated in Section 2.5 (Lines 315–316):“We propose increasing Mixup’s α value (compared to the baseline α used in full-cycle training) during the Enlightenment Period.”
> > > As indicated in the legend of Figure 4, the Alpha Boost strategy was not applied in this experiment, and the Mixup-α value remained consistent throughout the training process.
> > > Using Mixup-α=0.8 throughout training is a common configuration in ImageNet-related tasks (see Appendix Table 10).
> > > Either disabling Mixup entirely or increasing the Mixup-α value consistently across the whole training process would lead to performance degradation.
> > >
> > > 2. **In addition, it is unclear why increasing  reduces the number of zero-activations specifically during the enlightenment period and then increases again afterward. Would the number of zero-activations remain lower if the strategy were applied throughout the entire training process? Clarification or additional experiments on this point would be helpful.**
> > >
> > > Response:
> > > Please refer to Question 3 in the "Weaknesses".
> > > The Activation Revival Effect provides a novel explanation for why Mixup outperforms vanilla training, but it is only one of the optimization mechanisms of Mixup. The actual working mechanism is apparently more complex, and the optimal Mixup-α value in practice relies heavily on empirical results in previous studies. Additionally, Figure 4 shows that the most significant difference in the number of zero-activating neurons between Mixup-α = 1 and α = 2 occurs in the early training stage, while their values tend to converge in the middle and late stages.
> > >
> > >
> > > 3. **The paper provides ablation studies on data scale and pause epoch for the Mixup Pause strategy, but there is no ablation on model size. An additional analysis showing how model size affects the effectiveness of the Mixup Pause strategy would strengthen the experimental validation.**
> > >
> > > Response:
> > > As illustrated in Figure 5, Mixup Pause yields the most consistent and prominent performance gains for smaller models. This is fully consistent with the predictions of the mathematical model in our paper.
> > >
> > > 4. **A more thorough comparison with related work is needed. The paper should discuss prior theoretical studies on Mixup [1, 2, 3] as well as literature on the critical period [4, 5]. In particular, [1, 2] argue that Mixup should be applied only in the early phase of training, which directly contradicts the main claim of this paper. Meanwhile, [5] identifies a critical period around 50% training accuracy as crucial for preventing the loss of plasticity, which appears closely related to the authors’ notion of the enlightenment period. Providing a more detailed comparison that highlights these similarities and differences would significantly strengthen the paper’s positioning and clarify how this work advances beyond the existing studies.**
> > >
> > >
> > > Response:
> > >
> > > For prior theoretical studies, refer to Appendix A in detail, which includes a detailed discussion and related experiments on [1].
> > > Studies [1, 2]  focus on different temporal scales of the training process and emphasize the long-term over-regularization effect of Mixup, while our research on the "Enlightenment Period" focuses on the extremely early-stage training dynamics induced by Mixup. More importantly, our proposed strategy is complementary rather than contradictory to [1, 2]. Detailed analyses and experiments in Appendix A.2 (Lines 702–721) have demonstrated that combining the two types of strategies (regulating Mixup in the extremely early stage + stopping Mixup in the late stage) can achieve additive performance gains.
> > >
> > > For Study [3], the key difference is that it focuses on theoretical guarantees of Mixup for optimizing decision boundaries, while our paper centers on the unique training dynamics of Mixup during the enlightenment period. Study [4] explores the critical learning period of **linear networks**, while our work focuses on the critical early stage of training in DNNs(non-linear) (Appendix A.1 comprehensively summarizes previoust studies on early dynamics of DNNs). Study [5] focuses on the loss of plasticity in training restart scenarios, where the 50% accuracy serves as a test metric to evaluate the learning ability after restart; in contrast, the 50% accuracy in our paper defines a critical turning point for representation formation during training from scratch, which has an entirely different meaning.

---

### Author Response · Authors · 2025-11-18
**Urgent Concern Regarding Review Quality and Potential AI Reliance (Paper ID: [5986])**

Dear Area Chair,

We respectfully submit this formal concern regarding the reviews for our submission "Enlightenment Period Improving DNN Performance". While we welcome constructive criticism, we have identified alarming irregularities suggesting that **at least two reviewers (bMWW, iEY9) likely relied heavily on automated tools/AI without verifying the content and the expression is vague and ambiguous, making the review quite difficult to follow**. Other committed factual errors that indicate a failure to read the manuscript. These issues severely undermine the fairness of the review process.


- **Reviewer bMWW: Factual Confusion and Potential AI Reliance**
	Reviewer bMWW questions the 50% accuracy threshold for the Enlightenment Period, incorrectly asserting it applies to binary classification (where 50% is chance level). In reality, we explicitly tie the 50% threshold to experimental settings like ImageNet-1K (a 1000-class task), while binary classification is only used in our mathematical proofs.
	Of particular concern, Reviewer bMWW submitted 5 weaknesses and 3 questions, totaling 8 points. **In total, five of these eight items (across both weaknesses and questions) are merely different restatements of the same issue regarding the 50% accuracy threshold; moreover, they are expressed in a logically disorganized and difficult-to-follow manner, which makes it hard to believe that this review reflects a coherent line of human reasoning.**

- **Reviewer iEY9: False Technical Claim Suggesting Automated Parsing**
	Reviewer iEY9 claims Figures 1 and 7 are “low-resolution”(in weakness 5). This is factually impossible for a human reader to conclude because: All our figures are vector graphics (PDF), which remain infinitely sharp at any zoom level. However, automated PDF parsers (often used by AI summarizers to "read" papers) frequently fail to render vector graphics correctly, converting them to low-res bitmaps. A human reviewer viewing the PDF would inevitably see the vector quality.

- **Reviewer jpvs: Critical Factual Errors Contradicting the Text**
	Reviewer jpvs claims that our related work lacks “a more thorough comparison with The Benefits of Mixup for Feature Learning” (Question 4). . However, this comment ignores our explicit analysis and experiments in our related work(Lines 702–714): we not only discuss this paper in detail but also provide empirical evidence that combining our Enlightenment Period strategy with late-stage Mixup cessation yields additive performance gains. The reviewer’s oversight of this critical content strongly suggests a lack of thorough reading.


We do not dispute the need for rigorous review, but we request you re-evaluate these four reviewers’ comments to ensure they reflect a genuine engagement with our work. Fair assessment of our contributions (the Enlightenment Period framework and scenario-specific Mixup strategies) depends on reviews grounded in a complete reading of the manuscript.

Thank you for your time and attention to this matter.

Sincerely,

The Authors of “Enlightenment Period Improving DNN Performance”

---

### Meta-Review · Area_Chair_Booe · 2026-01-06

**Summary:**

The paper studies the use of Mixup during an early training phase termed the “Enlightenment Period” (ENP), defined as the period until training accuracy reaches approximately 50%, and proposes several ENP-specific strategies (Mixup Pause, Alpha Boost, High-Loss Removal). Reviewers generally found the paper clearly written and the topic interesting, with an intuitive framing of early training instability. However, there is broad agreement that the central conceptual definition of ENP is ill-posed and insufficiently justified, the theoretical analysis relies on oversimplified assumptions that do not convincingly transfer to modern deep networks, and the empirical gains are modest, fragile, and heavily dependent on additional hyperparameters.

**Reviewer Concerns:**

The rebuttal partially addressed secondary concerns, such as providing additional ImageNet-scale results for Alpha Boost, clarifying implementation details, and expanding explanations of the proposed Gradient Interference and Activation Revival effects. These additions help clarify the authors’ intended narrative and show that the methods can yield small, consistent gains in some controlled settings. However, major concerns remain unresolved. Multiple reviewers questioned the definition of ENP itself: using a fixed 50% accuracy threshold is arbitrary, task-dependent, and internally inconsistent across binary vs. multi-class classification, with no task-agnostic theoretical justification. The theoretical analysis is largely based on simplified linear or sigmoid models and orthogonality assumptions that do not realistically capture the optimization dynamics of deep CNNs or ViTs, weakening the link between theory and experiments.

**Reviewer Scores:**

All negative reviews.

---

### Decision · Program_Chairs · 2026-01-26

Reject